# 3DPhysVideo: 3D Scene Reconstruction and Physical Animation Leveraging a Video Generation Model via Consistency-Guided Flow SDE

## Abstract

Video generative models have made remarkable progress, yet they often yield visual artifacts that violate grounding in real-world physical dynamics. Recent works such as PhysGen3D tackle single image-to-3D physics through mesh reconstruction and Physically-Based Rendering, but challenges remain in modeling fluid dynamics and photorealism. This work introduces 3DPhysVideo, a novel training-free pipeline that generates physically realistic videos from a single image. We repurpose an off-the-shelf video model for two stages. First, we use it as a novel view synthesizer to reconstruct complete 360-degree 3D scene geometry by guiding the image-to-video (I2V) flow model with rendered point clouds derived from an initial 3D estimation. Second, after applying Material Point Method (MPM) physics simulation to this geometry, the simulated point cloud is used to guide the same I2V flow model to synthesize final, high-quality videos. Consistency-Guided Flow SDE, which decomposes the predicted velocity of the I2V flow model into denoising and consistency bias, allows us to effectively repurpose the model for both 3D reconstruction and simulation-guided video generation. Our method successfully bridges the gap from single-images to physically plausible videos while remaining efficient to run on a single consumer gpu. In the extensive experiments, our approach outperforms state-of-the-art baselines on both GPT-based evaluations and VideoPhy physics-consistency benchmark, across diverse scenarios including single-object, multi-object, and fluid interaction sequences.

## 1 Introduction

Recent advances in image-to-video (I2V) synthesis have achieved impressive fidelity of generated videos. These data-driven models (Blattmann et al., 2023; OpenAI, 2024; Google, 2025; Germanidis & Research, 2024), however, often lack fundamental understanding of real-world physics, resulting in implausible dynamics and photonics as demonstrated by several physics benchmarks (Motamed et al., 2025; Bansal et al., 2024; Meng et al., 2024). Several works aim to enhance the data-driven video generation models with physics awareness for general scenes from single images. Force Prompting (Gillman et al., 2025) exploits physics-informed data to induce physics interactions into its model, while VLIPP (Yang et al., 2025b) employs vision-language models (VLM) for physics reasoning prior to data-driven video generation. However, these approaches break down in out-of-domain physics scenarios due to their reliance on data-driven video models.

To achieve better generalization and accurate physics modeling, another line of works integrate explicit physical simulation into 3D Gaussian Splatting (Kerbl et al., 2023) by directly applying simulation results (Xie et al., 2024) or learning differentiable simulators (Zhang et al., 2024; Huang et al., 2025; Liu et al., 2025) with video model priors. These physics-equipped Gaussians are then rendered to generate physically plausible videos. The 3D GS-based methods typically require multi-view inputs, which are difficult to obtain in practical settings. Recent attempts combine simulation accuracy with improved visual synthesis. PhysMotion (Tan et al., 2024) reconstructs 3D GS using LGM (Tang et al., 2024) before applying simulation and refining the rendered output with video models. PhysGen3D (Chen et al., 2025a), which extends PhysGen (Liu et al., 2024) to 3D, mod-

els 3D physics from a single image using mesh reconstruction and Physically-Based Rendering (PBR). Both approaches, however, rely on object-centric 3D reconstruction models, and the mesh representation in PhysGen3D leads to fabric-like artifacts and limitations in fluid dynamics where maintaining vertex connectivity is challenging.

In this work, we propose a novel pipeline, named 3DPHYSVIDEO, for single image-to-3D dynamics video generation, via two non-trivial stages: single-view 3D reconstruction and simulation-guided video generation. We propose *Consistency-Guided Flow SDE*, which decomposes the predicted velocity of I2V flow model into denoising bias and consistency bias, repurposing the I2V model as both a novel-view synthesizer and a simulation-guided video generation model with its reinforced consistency. As a novel-view synthesizer, the I2V model generates a 360-degree orbit video with geometry guidance obtained by unprojecting a single-view to the world coordinate using a point cloud reconstruction model (Wang et al., 2025b). Since the orbit video obtained by Consistency-Guided Flow SDE is inherently world-consistent, we then obtain 3D scene geometry by simply unprojecting it using the same feedforward model. Subsequently, using the I2V model as a simulation-guided video generator, we generate the high quality output videos following 3D physics dynamic of various materials including solids, fluids, and viscous substances simulated through MPM. 3DPHYSVIDEO (1) obtains 3D geometry of more generic scenes not relying on any object-centric 3D data-driven models, (2) offers high controllability by allowing users to specify desired conditions such as velocity, mass, and material properties in 3D space through physics simulation, and (3) generates photorealistic videos that follow this simulated physics dynamics. We validate our approach through extensive experiments, both quantitatively via human evaluation, GPT-scores and VideoPhy (Bansal et al., 2024) benchmarks, and qualitatively, in comparison to state-of-the-art data-driven video generation models (e.g., Gen-3, Sora, VEO), the most-relevant work PhysGen3D, physics-aware video model VLIPP, and MotionClone (Ling et al., 2025), Go-with-the-Flow (Burgert et al., 2025), MagicMotion (Li et al., 2025) as well as ablation studies on the effectiveness of our consistency-guided SDE. Results show that our method significantly improves physical realism and semantic consistency while maintaining competitive photorealism, particularly in challenging multi-object and fluid interaction scenarios. Our major contributions are:

- We propose 3DPHYSVIDEO, a novel training-free pipeline that combines pre-trained I2V model priors with physics simulation to generate 3D physically plausible videos from a single image. The I2V model serves as both a novel-view synthesizer and a simulation-guided video generation model.

- We propose Consistency-Guided Flow SDE, a novel method that decomposes the predicted velocity of I2V flow model into denoising bias and consistency bias, yielding videos that are consistent with input and guided images, while maintaining the photorealism.

- Extensive experiments demonstrate that our pipeline generates videos with superior physical and photo realism compared to state-of-the-art methods across diverse scenes, while providing flexible user-intended generation.

## 2 RELATED WORK

**3D Physics Dynamics.** 3D Gaussian Splatting (3D GS) (Kerbl et al., 2023) has been widely adopted to represent 3D scenes. PhysGaussian (Xie et al., 2024) and subsequent works (Huang et al., 2025; Zhang et al., 2024; Cai et al., 2024; Liu et al., 2025; Lin et al., 2025; Mittal et al., 2025) have integrated physics modeling using the Material Point Method (MPM) (Stomakhin et al., 2013; Jiang et al., 2016) to assign physical properties to each Gaussian particle. Recent advances in single-image 3D reconstruction have enabled recovering structure from a single view. Some methods directly predict 3D shapes (Xu et al., 2024), while others adopt a two-step approach: synthesizing multi-view images (Liu et al., 2023; Shi et al., 2024) then reconstructing geometry (Tang et al., 2024; Chen et al., 2025b). Beyond object-centric scenarios, several studies (Wang et al., 2025b; Yu et al., 2025a;b) extend to scene-level reconstruction. Building on these advances in single-view 3D reconstruction, several works (Chen et al., 2025a; Tan et al., 2024) have enabled physics simulation directly from single images using MPM-based approaches combined with visual synthesis models. These methods reveal the feasibility of generating physically realistic dynamic scenes directly from single-view inputs.

**Motion Conditioned Video Generation.** A wide variety of methods have been explored for controllable video generation. Optical flow, the most widely used motion representation or related motion

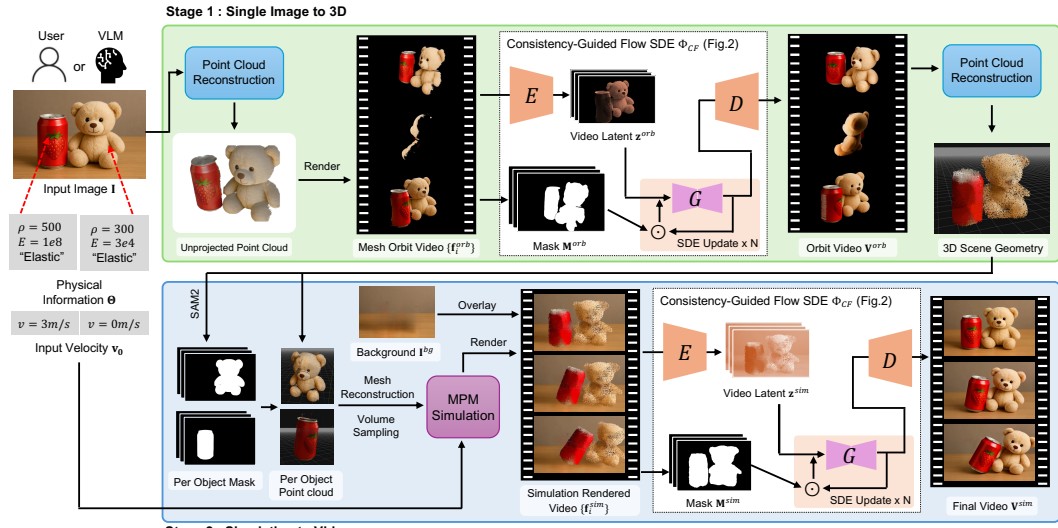

Figure 1: **Overall Pipeline.** Starting from a single image, 3DPHYSVIDEO reconstructs 3D scene geometry via 360-degree orbit video synthesis, then conducts particle-based physics simulation to produce photorealistic videos from the simulation results.

fields, has been employed in methods (Ni et al., 2023; Liang et al., 2024; Chen et al., 2023; Burgert et al., 2025), as conditioning signals, enabling temporal coherence and global motion controls but often lacking fine physical details. On the progress of point tracking models (Xiao et al., 2024; 2025), another line of work explores trajectory and point-based control (Geng et al., 2025; Jeong et al., 2025), which flexibly handles both dense and sparse point conditions. Region and entity-based approaches (Wu et al., 2024; Li et al., 2025; Haonan Qiu et al., 2024) provide fine-grained controls via masks, bounding boxes, or landmarks, though they tend to overlook high-frequency cues from simulation. Unified controllers such as (Zhang et al., 2025; Wang et al., 2024) aim to combine heterogeneous signals within a single framework. In parallel, training-free paradigms (Jain et al., 2024; Ling et al., 2025) demonstrate the feasibility of repurposing pretrained video diffusion models without additional training. Camera control and 3D-aware methods have also gained attention, including (He et al., 2025; Yang et al., 2024; Wu et al., 2025) which improve multi-view consistency and enable explicit camera trajectory conditioning. Beyond the trajectory or mask-based conditioning, the recent work on force prompting (Gillman et al., 2025) introduces physical forces as control signals for video generation, enabling both localized point interactions and global effects. This approach demonstrates that video models can generalize intuitive force-based dynamics from limited synthetic training, highlighting an alternative paradigm for incorporating physically meaningful controls.

**Flow-based Editing.** Flow matching (Lipman et al., 2023) has demonstrated computational advantages through straight-line ordinary differential equation (ODE) trajectories over diffusion models. Recent works have explored various editing applications including text-driven image or video editing through inversion methods using score distillation (Yang et al., 2025a), rectified stochastic differential equation (SDE) (Song et al., 2021; Rout et al., 2025), Taylor expansion solvers (Wang et al., 2025a), and predictor-corrector frameworks (Jiao et al., 2025). However, editing for image-to-video flow matching models that leverages consistency with input images remains underexplored.

## 3 3DPHYSVIDEO

Given a single RGB image $\mathbf{I} \in \mathbb{R}^{H \times W \times 3}$, our goal is to generate a physically plausible video $\mathbf{V}^{\text{sim}}$ through physics simulation. Our pipeline is designed to fully exploit the rich prior of a video generation model $G$ (Zhang & Agrawala, 2025). The first stage repurposes $G$ as a novel view synthesizer, producing an orbit video $\mathbf{V}^{\text{orb}}$ and corresponding 3D geometry from the input image $\mathbf{I}$ (Sec. 3.1). We then run physics simulation based on the 3D reconstruction, yielding simulated point trajectories $\mathcal{P} = \{\mathbf{P}_i\}_{i=1}^{L}$. In the second stage, we map these raw simulated trajectories $\mathcal{P}$ into a photorealistic and temporally coherent final video $\mathbf{V}^{\text{sim}}$ (Sec. 3.2). Both stages are done by our novel Consistency-Guided Flow SDE, $\Phi_{\text{CF}}$, introduced in (Sec. 3.3).

## 3.1 STAGE 1: SINGLE IMAGE TO 3D

Obtaining 3D geometry from a single image is fundamentally ill-posed, due to regions invisible from the input view. Hence, it is critical to exploit the strong priors of generative models. Unlike previous works (Chen et al., 2025a) that rely on object-level reconstruction (Xu et al., 2024), which individually process each object and therefore discard inter-object information such as relative poses and spatial relationships, we instead leverage a generic video generation model G (Zhang & Agrawala, 2025) to reconstruct the entire scene jointly. Throughout our pipeline, all necessary masks are obtained using SAM2 (Ravi et al., 2025).

**Point Cloud Unprojection.**    Given the image $\mathbf{I}$, we unproject a point cloud into the world coordinate system using a point cloud reconstruction model (Wang et al., 2025b). To retain only foreground points of interest for physics modeling, we apply the foreground mask of $\mathbf{I}$ to remove background points.

**Mesh Orbit Rendering.**    As shown in Fig. 1, the unprojected point cloud provides correct geometry for visible regions from the input view. Therefore, to utilize it as a geometry guidance for generating a world-consistent orbit video, we convert it to a mesh to fill gaps between sparse points and render it by moving the camera along a 360-degree orbit trajectory. This produces a mesh orbit video $\{\mathbf{f}_i^{orb}\}_{i=1}^{K}$ in black background, which yet has a significant number of missing pixels due to occlusions.

**World-Consistent Orbit Video Generation.**    To obtain a world-consistent orbit video $\mathbf{V}^{\text{orb}}$ following the geometry guidance of $\{\mathbf{f}_i^{orb}\}_{i=1}^{K}$, we repurpose the video model $G$ as a novel-view synthesizer. We achieve this using our Consistency-Guided Flow SDE, $\Phi_{\text{CF}}$ (Sec. 3.3) along with masking and inversion strategies (Mokady et al., 2023; Wang et al., 2025a).

$$\mathbf{V}^{\text{orb}} = \Phi_{\text{CF}}(\{\mathbf{f}_i^{orb}\}_{i=1}^{K}, \mathbf{I}, \mathbf{M}^{\text{orb}}; G). \tag{1}$$

Where $\mathbf{M}^{orb}$ is the video mask obtained by concatenating the per-frame masks of $\{\mathbf{f}_i^{\text{orb}}\}_{i=1}^{K}$. This mask helps preserve the geometry guidance regions, while enabling the use of video model $G$ that iteratively fills the empty regions to be semantically consistent with the input image $\mathbf{I}$.

**3D Scene Geometry.**    Since $\mathbf{V}^{\text{orb}}$ is inherently world-consistent by following the geometry guidance, we obtain 3D scene geometry by simply unprojecting it back to the world coordinate using the same point cloud reconstruction model, without requiring any object-centric 3D reconstruction models as in prior-arts. This scene-level strategy preserves inter-object relationships during reconstruction, such as relative poses and spatial relationships, which are absent in object-centric approaches.

## 3.2 STAGE 2: SIMULATION TO VIDEO

**Per-object Segmentation & Post-processing.**    The point cloud from Sec. 3.1 captures the scene structure but lacks explicit object separation and interior points needed for simulation. We get per-object segment masks and apply it on the scene reconstruction to yield per-object point clouds. Then, we remove outliers, reconstruct watertight meshes via Poisson reconstruction (Kazhdan et al., 2006), and sample volumetric points to form MPM-ready point clouds $\{\mathbf{Q}_i^o\}$ with colors interpolated from the input point cloud. We also detect the ground plane $\pi$ via RANSAC (Fischler & Bolles, 1981). Note that mesh reconstruction serves only for volumetric point sampling, not 3D representation. Point cloud and mesh processing details are provided in the supplementary material.

**MPM Simulation & Rendering.**    Following (Chen et al., 2025a), we obtain the initial velocity $\mathbf{v}_0$ and physical properties $\Theta$ through either automatic inference using GPT-5 or direct user specification, including material-specific parameters such as object elasticity, density, and surface friction coefficients required for realistic physics simulation. With the obtained object-level point clouds $\{\mathbf{Q}_i^o\}$ and parameters $\pi$, $\mathbf{v}_0$ and $\Theta$, we run an MPM simulator (Jiang et al., 2016; Hu et al., 2019) to generate simulated output $\mathcal{P} = \text{MPM}(\{\mathbf{Q}_i^o\}, \pi, \mathbf{v}_0, \Theta)$, where $\mathcal{P} = \{\mathbf{P}_i\}_{i=1}^{L}$ represents point trajectories that captures physically accurate deformations and motions. We then render the raw simulated trajectories $\mathcal{P}$ using a point cloud renderer (Meta, 2024), preserving the color information throughout the simulation. These rendered frames are then overlaid onto the inpainted background image $\mathbf{I}^{\text{bg}}$, yielding the simulation-rendered video $\{\mathbf{f}_i^{\text{sim}}\}_{i=1}^{L}$.

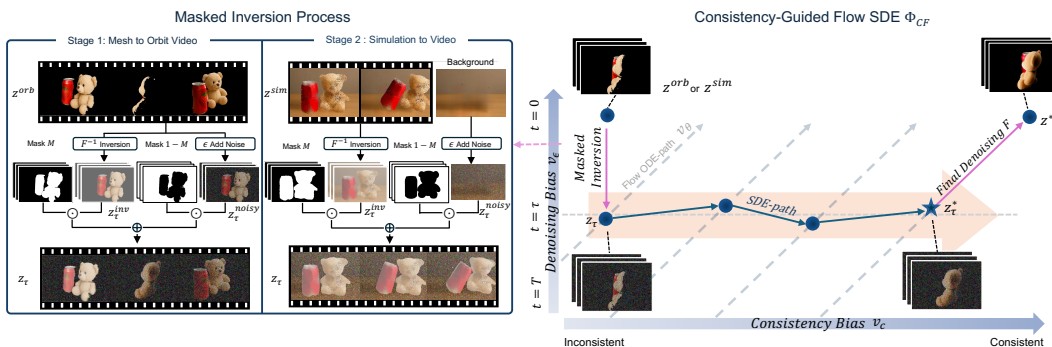

Figure 2: **Consistency-Guided Flow SDE $\Phi_{\mathbf{CF}}$**. The masked inversion process (left) combines inverted and noisy latents to obtain the starting latent for optimization. The iterative consistency optimization (right) via consistency-guided flow SDE refines this latent to produce semantically or photorealistically consistency-optimal videos. The blue arrow trajectory from $\mathbf{z}_\tau$ to $\mathbf{z}_\tau^*$ illustrates our SDE optimization path.

**Photorealistic Video Generation.** The rendered video $\{\mathbf{f}_i^{sim}\}_{i=1}^L$ shows physically accurate deformations and motions. Since it is rendered from points, however, it lacks photorealism with missing shadows and lighting effects. To address this, we now repurpose the video model $G$ as a simulation-guided video generation model using $\Phi_{\text{CF}}$ with the masking and inversion strategies, similar to Sec. 3.1.

$$\mathbf{V}^{\text{sim}} = \Phi_{\text{CF}}(\{\mathbf{f}_i^{sim}\}_{i=1}^L, \mathbf{I}, \mathbf{I}^{\text{bg}}, \mathbf{M}^{\text{sim}}; G). \tag{2}$$

Where $\mathbf{M}^{\text{sim}}$ is the video mask obtained by concatenating the per-frame masks of $\{\mathbf{f}_i^{sim}\}_{i=1}^L$. This mask with the background image enforces adherence to the simulated motion guidance regions and allows the video model $G$ to iteratively refine the appearance for consistency with the input image $\mathbf{I}$. As a result, the final video $\mathbf{V}^{\text{sim}}$ is both physically faithful to the simulated dynamics and photorealistic.

### 3.3 CONSISTENCY-GUIDED FLOW SDE $\Phi_{\text{CF}}$

A key contribution of our framework lies in leveraging the consistency-guided Flow SDE $\Phi_{\text{CF}}$ across two distinct stages. In Sec. 3.1, $\Phi_{\text{CF}}$ yields a world-consistent orbit video from an incomplete mesh orbit video. In Sec. 3.2, $\Phi_{\text{CF}}$ translates simulation-rendered video into a photorealistic video. By reusing the same mechanism with different guidances (geometry in the first stage and physics simulation in the second), our method unifies geometry reconstruction and physics-based animation under a single coherent framework.

**Masked Inversion Strategy.** From Eq. 1 and Eq. 2, we first encode the inputs of $\Phi_{\text{CF}}$ - the $\{\mathbf{f}_i^{\text{orb}}\}_{i=1}^K$ or $\{\mathbf{f}_i^{\text{sim}}\}_{i=1}^L$, $\mathbf{I}$, and $\mathbf{I}^{\text{bg}}$ - to obtain $\mathbf{z}^{\text{orb}}$ or $\mathbf{z}^{\text{sim}}$, $\mathbf{z}^I$, and $\mathbf{z}^{bg}$ respectively. Then, as shown in Fig. 2, we obtain two intermediate latents at a diffusion step $t = \tau$: $\mathbf{z}_\tau^{\text{inv}}$ through the inversion process on $\mathbf{z}^{\text{orb}}$ or $\mathbf{z}^{\text{sim}}$, and $\mathbf{z}_\tau^{\text{noisy}}$ through the forward process on $\mathbf{z}^{\text{orb}}$ or $\mathbf{z}^{bg}$ for the two stages respectively. Finally, we obtain the combined video latent $\mathbf{z}_\tau = \mathbf{M} \cdot \mathbf{z}_\tau^{\text{inv}} + (1 - \mathbf{M}) \cdot \mathbf{z}_\tau^{\text{noisy}}$, where $\mathbf{M}$ is the video mask, $\mathbf{M}^{\text{orb}}$ or $\mathbf{M}^{\text{sim}}$. This masked inversion strategy allows $\mathbf{z}_\tau$ to preserve the guidance regions corresponding to $\mathbf{M}$ through the inversion process while enabling the use of video model $G$ through the intermediate latent space at $\tau$.

**Consistency-Guided Flow Stochastic Differential Equation (SDE).** The velocity prediction model $v_\theta$ of the I2V flow model $G$ (Lipman et al., 2023; Zhang & Agrawala, 2025) acquires a consistency bias through training to guide a intermediate video latent $\mathbf{z}_t$ toward consistency with input image. Hence, a straightforward approach to obtain a video more semantically or photorealistically consistent with $\mathbf{z}_I$ from $\mathbf{z}_\tau$ is to follow the standard generation process using the velocity predicted by $v_\theta$. However, since $v_\theta$ inevitably possesses both consistency bias and denoising bias, we empirically verified that the generation process guided by both biases produces consistency-suboptimal videos (see Fig. 7). To achieve consistency-optimal video generation, we decompose $v_\theta$ into a velocity prediction model $v_c$ that captures the consistency bias and a velocity prediction

model $v_\epsilon$ that captures the denoising bias:

$$v_\theta(\mathbf{z}_t, \mathbf{z}_I, t) = v_c(\mathbf{z}_t, \mathbf{z}_I, t) + v_\epsilon(\mathbf{z}_t, \mathbf{z}_I, t). \tag{3}$$

Specifically, our goal is to leverage $v_c$ at the intermediate diffusion step $t = \tau$ to optimize the video latent $\mathbf{z}_\tau$ into a consistency-optimal video latent $\mathbf{z}_\tau^*$ in Fig. 2. By leveraging $v_\epsilon$ to maintain the original distribution $q = \mathcal{N}((1 - \tau)\boldsymbol{\mu}, \tau^2 \mathbf{I})$ at $\tau$, we ensure that the enforced consistency does not lean toward high-frequency details ($t \approx 0$) and low-frequency components ($t \approx T$). Accordingly, our objective is to find the target distribution $p^*$ that enables sampling $\mathbf{z}_\tau^*$ which maximize $C(\cdot, \mathbf{z}_I)$ (the consistency with $\mathbf{z}_I$) while minimizing KL divergence (Kullback & Leibler, 1951) with $q$, balanced by a regularization parameter $\beta$:

$$p^* = \arg\max_p \ \mathbb{E}_{\mathbf{z}_\tau \sim p}\left[C(\mathbf{z}_\tau, \mathbf{z}_I)\right] - \frac{1}{\beta}\mathbb{D}_{\mathrm{KL}}(p \,\|\, q). \tag{4}$$

To solve this optimization, the consistency bias model $v_c$ approximates $\nabla C(\mathbf{z}_\tau, \mathbf{z}_I)$ to maximize the first term, while the denoising bias model $v_\epsilon$ is utilized to approximate the score function $\nabla_{\mathbf{z}_\tau} \log q(\mathbf{z}_\tau)$ for the KL divergence term. Using these approximated gradients, we formulate an overdamped Langevin stochastic differential equation (SDE) (Risken, 1989; Song et al., 2021) to sample from the optimal distribution $p^*$ and discretize it via Euler-Maruyama method (Kloeden & Pearson, 1977; Gianfelici, 2008). Through this process with $v_c = v_\theta - v_\epsilon$ from our decomposition, we verify that at the specific choice $\beta = \frac{1-\tau}{\tau}$, the $v_\epsilon$ term completely cancels out. This enables us to achieve our consistency optimization objective using only the known $v_\theta$:

$$\mathbf{z}_\tau^{(n+1)} = (1 - \frac{\gamma}{\tau})\mathbf{z}_\tau^{(n)} + \frac{1-\tau}{\tau}\gamma \, v_\theta(\mathbf{z}_\tau^{(n)}, \mathbf{z}_I, \tau) + \sqrt{2\gamma} \, \boldsymbol{\epsilon}^{(n)}, \quad \boldsymbol{\epsilon}^{(n)} \sim \mathcal{N}(0, \mathbf{I}). \tag{5}$$

Where $\gamma$ denotes the step size of discretization. Through $N$ iterations of this final iterative formula on $\mathbf{z}_\tau$, we obtain $\mathbf{z}_\tau^*$ semantically or photorealistically consistent with $\mathbf{z}_I$. In this process, we mask $\mathbf{z}_\tau^{(n+1)}$ with the $\mathbf{M}$, to maintain the guidance regions of $\mathbf{z}_\tau$. Finally, we obtain $\mathbf{V}^{\mathrm{orb}}$ or $\mathbf{V}^{\mathrm{sim}} = D(F(\mathbf{z}_\tau^*))$ through the final denoising process $F$ and the decoding process $D$. The detailed derivation and algorithm are provided in the supplementary material.

## 4 EXPERIMENTS

**Implementation Details.** In our proposed pipeline, we use Framepack (Zhang & Agrawala, 2025), an auto-regressive DiT-based I2V flow model. We set $\tau$ to the value at step 20 out of the 25 total inference steps, and we use $N = 10$ and $\gamma = 0.2$ for SDE optimization. All experiments were conducted on a single NVIDIA RTX 3090 GPU. Analysis of the SDE hyperparameters and the pipeline's inference time breakdown are provided in Sec. F and Sec. G of the supplementary material, respectively.

**Data Sets.** We curate a set of 10 diverse scenes drawn from multiple sources, including Phys-Gen3D (Chen et al., 2025a), the Physics-IQ benchmark (Motamed et al., 2025), internet-sourced examples, and GPT-generated scenarios. The dataset spans three categories: **fluid**, **single-object**, and **multi-objects**. This design covers a wide spectrum of physical interactions, from simple object dynamics to complex fluid and multi-body interactions.

**Compared Methods.** We evaluate our full pipeline (Stage 1 + Stage 2) against several strong baselines. FramePack (Zhang & Agrawala, 2025) serves as a direct baseline to isolate the effect of our physics-guided framework. Sora (OpenAI, 2024), Gen-3 Alpha (Germanidis & Research, 2024), and Veo-3 (Google, 2025) are state-of-the-art commercial video generation models, representing the strongest purely data-driven approaches. We further include PhysGen3D (Chen et al., 2025a), the most relevant prior work on single image-to-3D physics-based video generation. We also compare with VLIPP (Yang et al., 2025b), a representative of methods that adapt pretrained video generation models (e.g., Force Prompting (Gillman et al., 2025)) to physically plausible motions without explicit simulations. This part of comparisons are to methods that are applicable for the setting of single-image input, which is the central focus of our work. The approaches of physics-integrated Gaussian Splatting (Xie et al., 2024; Cai et al., 2024; Lin et al., 2025; Mittal et al., 2025) are not included, as they require multi-view inputs. For Stage 2 (simulation to video), we compare against MotionClone (Ling et al., 2025), Go-with-the-Flow (Burgert et al., 2025), and MagicMotion (Li et al., 2025), which represent motion-prior, flow-based, and mask-based conditioning, respectively. This highlights the strength of our consistency-guided Flow SDE in preserving input appearance while following simulated dynamics.

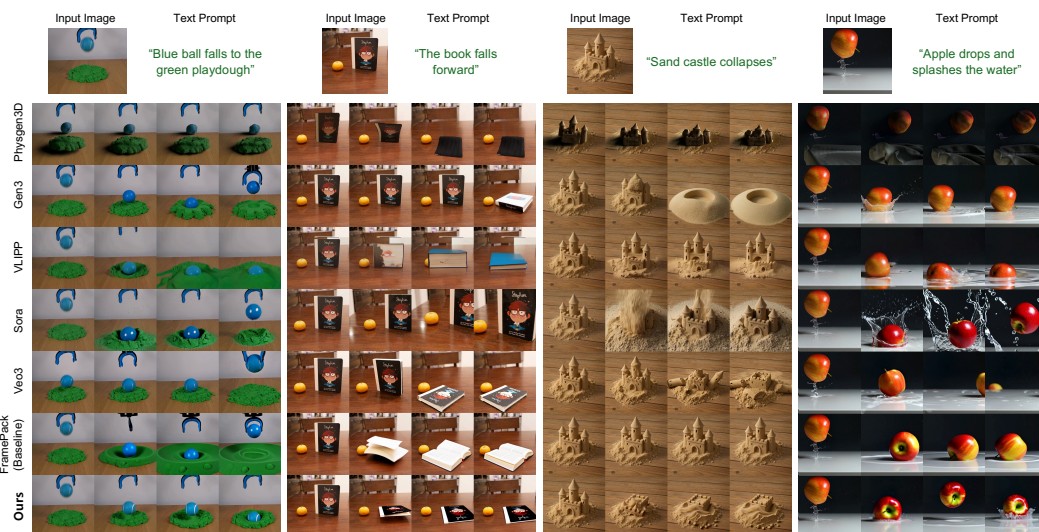

Figure 3: Qualitative comparison across four representative scenarios. From left to right: (i) the ball drop scene from the Physics-IQ benchmark, (ii) the book-fall scene from the PhysGen3D dataset, (iii) a synthetic sand-castle collapse scene self-made, and (iv) an apple falling scene from internet.

**Qualitative Comparison with SOTA Video Models.** We provide qualitative results across four scenarios: the ball-drop scene from Physics-IQ, the book-fall scene from PhysGen3D, our synthetic sand-castle collapse scene, and apple dropping on water scene from internet (Fig. 3). Across all settings, the prior models often produce artifacts such as unnatural rebounds, missing object details, or unstable backgrounds. Specifically, PhysGen3D relies on object-centric 3D reconstruction, which discards relative poses and spatial relationships as shown in the first scenario, and mesh-based PBR rendering, often leading to distorted textures and limitations in fluid dynamics as observed in the fourth scenario. General-purpose video generation models (Sora, Gen-3, Veo) often produce physically implausible behaviors, such as objects suddenly disappearing, appearing, or moving unnaturally. Recent physics reasoning approaches using VLM chain-of-thought, such as VLIPP show improved physical plausibility, but without explicit physics simulation they still exhibit clear limitations. In contrast, our method consistently preserves scene appearance while generating physically plausible object motions and realistic interactions with the environment.

Table 1: Comparison results of GPT-5 evaluation (PhysReal, PhotoReal, Semantic), VideoPhy Physical Commonsense Score (VPhy), and Human evaluation (PhysReal, PhotoReal, Semantic) across different scenario groups.

| | One Object | | | | | | | Liquid | | | | | | |
| Method | PhysR | GPT PhotoR | Sem | VPhy | Human PhysR | PhotoR | Sem | PhysR | GPT PhotoR | Sem | VPhy | Human PhysR | PhotoR | Sem |
|---|---|---|---|---|---|---|---|---|---|---|---|---|---|---|
| PhysGen3D | 0.206 | 0.251 | 0.317 | 0.200 | 1.47 | 1.60 | 2.05 | 0.119 | 0.205 | 0.118 | 0.037 | 1.45 | 1.66 | 1.52 |
| Gen3 | 0.506 | 0.801 | 0.428 | 0.348 | 2.78 | 3.25 | 3.32 | 0.513 | 0.845 | 0.339 | 0.126 | 3.08 | 3.68 | 2.70 |
| VLIPP | 0.426 | 0.425 | 0.425 | 0.372 | 1.75 | 2.43 | 2.35 | 0.544 | 0.792 | 0.574 | **0.192** | 2.14 | 2.92 | 2.71 |
| Sora | 0.306 | 0.666 | 0.107 | 0.257 | 1.95 | 2.58 | 1.80 | **0.722** | 0.852 | 0.599 | 0.073 | 1.82 | 3.05 | 2.62 |
| VEO | **0.786** | 0.871 | **0.837** | 0.223 | **3.25** | **3.90** | 3.60 | 0.689 | **0.867** | 0.656 | 0.098 | 3.40 | 3.78 | 3.65 |
| Framepack(Baseline) | 0.484 | **0.892** | 0.488 | 0.440 | 2.12 | 2.90 | 1.62 | 0.668 | 0.851 | 0.640 | 0.101 | 2.47 | 3.50 | 3.08 |
| **Ours** | 0.726 | 0.880 | 0.656 | **0.510** | 3.15 | 3.62 | **3.73** | 0.702 | 0.843 | **0.764** | 0.120 | **3.65** | **3.90** | **4.17** |
| | Multi Object | | | | | | | Overall | | | | | | |
| Method | PhysR | GPT PhotoR | Sem | VPhy | Human PhysR | PhotoR | Sem | PhysR | GPT PhotoR | Sem | VPhy | Human PhysR | PhotoR | Sem |
| PhysGen3D | 0.183 | 0.535 | 0.191 | **0.193** | 1.43 | 1.63 | 1.61 | 0.171 | 0.360 | 0.206 | 0.149 | 1.45 | 1.63 | 1.71 |
| Gen3 | 0.554 | 0.811 | 0.582 | 0.124 | 2.25 | 3.09 | 2.75 | 0.528 | 0.818 | 0.469 | 0.192 | 2.66 | 3.32 | 2.91 |
| VLIPP | 0.662 | 0.780 | 0.740 | 0.186 | 2.05 | 2.76 | 2.59 | 0.561 | 0.682 | 0.603 | 0.244 | 1.98 | 2.71 | 2.55 |
| Sora | 0.703 | 0.767 | **0.762** | 0.102 | 2.16 | 2.78 | 2.66 | 0.595 | 0.762 | 0.528 | 0.140 | 1.99 | 2.80 | 2.39 |
| VEO | 0.662 | **0.874** | 0.499 | 0.083 | 2.85 | 3.49 | 3.29 | 0.705 | **0.871** | 0.641 | 0.129 | 3.14 | 3.70 | 3.49 |
| Framepack(Baseline) | 0.308 | 0.806 | 0.262 | 0.147 | 2.01 | 3.09 | 2.43 | 0.461 | 0.844 | 0.434 | 0.221 | 2.18 | 3.16 | 2.38 |
| **Ours** | **0.721** | 0.840 | 0.756 | 0.191 | **3.46** | **3.77** | **3.95** | **0.717** | 0.852 | 0.730 | **0.266** | **3.43** | **3.76** | **3.95** |

**GPT-based Evaluation.** We adopt GPT-based assessments to evaluate *Physical Realism*, *Photorealism*, and *Semantic Consistency*. Physical Realism measures whether the generated motion follows

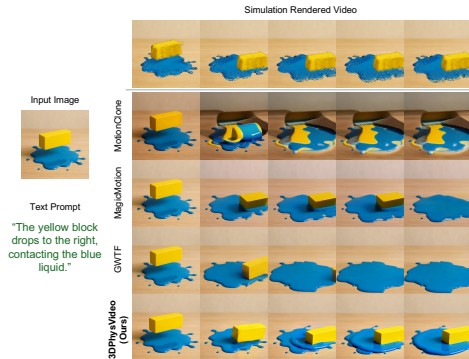

Figure 4: Qualitative comparison on Simulation to Video.

Table 2: GPT-5 evaluation results of Stage 2: Simulation to Video.

| Method | PhysReal | PhotoReal | Semantic |
|---|---|---|---|
| MotionClone | $0.396_{\pm 0.043}$ | $0.631_{\pm 0.060}$ | $0.387_{\pm 0.116}$ |
| MagicMotion | $0.445_{\pm 0.073}$ | $0.575_{\pm 0.042}$ | $0.573_{\pm 0.052}$ |
| Go-with-the-Flow | $0.632_{\pm 0.064}$ | $0.812_{\pm 0.043}$ | $0.689_{\pm 0.052}$ |
| **Ours** | $\mathbf{0.711_{\pm 0.061}}$ | $\mathbf{0.862_{\pm 0.020}}$ | $\mathbf{0.758_{\pm 0.057}}$ |

Table 3: Vbench evaluation results of Stage 2 : Simulation to Video.

| Method | AQ | BC | IQ | MS | SC | TF |
|---|---|---|---|---|---|---|
| MotionClone | 0.551 | 0.865 | 61.159 | 0.980 | 0.769 | 0.972 |
| MagicMotion | 0.531 | 0.921 | 60.819 | 0.990 | 0.877 | 0.986 |
| Go-with-the-Flow | 0.540 | 0.940 | 62.164 | 0.995 | 0.909 | 0.992 |
| **Ours** | **0.580** | **0.942** | **63.532** | **0.996** | **0.916** | **0.993** |

physical laws such as gravity, elasticity, and collisions, while Photorealism assesses the fidelity of generated frames, including texture, lighting, and rendering quality. Semantic Consistency evaluates how well generated motions correspond to intended trajectories or simulated dynamics. Following PhysGen3D (Chen et al., 2025a), we provide GPT with the input image, the motion context, and sampled video frames, and collect scores across the three axes. Details on GPT-based evaluation protocol can be found in supplementary material.

Tab. 1 summarizes the results. Our method achieves the highest overall scores in physical realism and semantic consistency, noting our core objective of enhancing physical plausibility in generated videos. Our score on photorealism remains competitive with state-of-the-art methods, showing that improved physical plausibility does not come at the cost of visual quality.

Our approach performs particularly well in multi-object and liquid scenarios, outperforming baselines in settings with complex physical interactions. This aligns with our design choice of incorporating explicit physics simulation: whereas existing methods often suffer from object hallucination, such as objects disappearing, multiplying, or moving unnaturally, our approach maintains consistent object identities and interactions throughout the sequences. The improvements in the multi-object scenarios demonstrate the advantage of grounding video generation in physical simulation for handling complex interactions that pure generative models struggle with.

**Human Evaluation.** We additionally conduct human evaluation with 20 participants on Physical Realism, Photorealism, and Semantic Consistency across all scenarios. Human evaluation protocol is mainly borrowed from (Chen et al., 2025a), as more details are provided in the supplementary material. As shown in Tab. 1, our method consistently receives the highest ratings from human judges, particularly in multi-object and liquid scenarios. Compared to baselines, participants judged our results as both more physically plausible and semantically faithful, while maintaining strong photorealism.

**Physics Consistency Evaluation.** To further assess the physical plausibility of generated videos, we evaluate using the VideoPhy benchmark (Bansal et al., 2024), which measures adherence to intuitive physical laws. Results in the **VPhy** columns of Tab. 1 show our method achieves the highest overall score with strong robustness across physical settings, maintaining stable interactions and realistic physical behavior without compromising visual quality. While PhysGen3D shows strong performance in the multi-object settings, it encounters significant challenges in fluid scenarios, leading to an imbalanced performance across different scene types.

**Comparison with SOTA Methods for Stage 2: Simulation to Video.** Tab. 2 and Tab. 3 reports GPT-based evaluation and VBench (Huang et al., 2024) results for Stage 2, where simulation outputs are converted into realistic videos via our physics-guided pipeline or other methods. The same simulation obtained by the proposed method stage 1 is given to all methods. As in the previous evaluation, Physical Realism (PhysReal), Photorealism (PhotoReal), and Semantic Consistency (Semantic) are assessed by GPT-5, while the additional metrics (AQ, BC, IQ, MS, SC, TF) are drawn from the VBench benchmark for video quality. Our approach achieves the best performance across all metrics, demonstrating not only superior physical plausibility but also consistently strong visual fidelity.

**Qualitative Comparison for Simulation-to-Video.** We further evaluate our method on a scenario where a yellow block falls to the right and contacts blue liquid (Fig. 4). For this experiment, all methods are given the same inputs: the simulation-rendered video, the original input image, and the text prompt describing the scene. Fig. 4 shows that the different methods exhibit characteristic behaviors under this setting. MotionClone (Ling et al., 2025) generates unstable object representations, where the yellow block loses its shape and blends with the liquid. Go with the Flow (Burgert et al., 2025) often deviates from the simulated dynamics, producing a sliding trajectory that continues beyond the frame instead of halting at the point of impact. MagicMotion (Li et al., 2025) broadly follows the simulated trajectory but occasionally causes the yellow block to vanish beneath the liquid, a limitation likely stemming from mask-based conditioning without explicit object boundary constraints. In contrast, our method preserves the block's geometry and texture while accurately reproducing the simulated dynamics, leading to realistic and coherent fluid–object interactions.

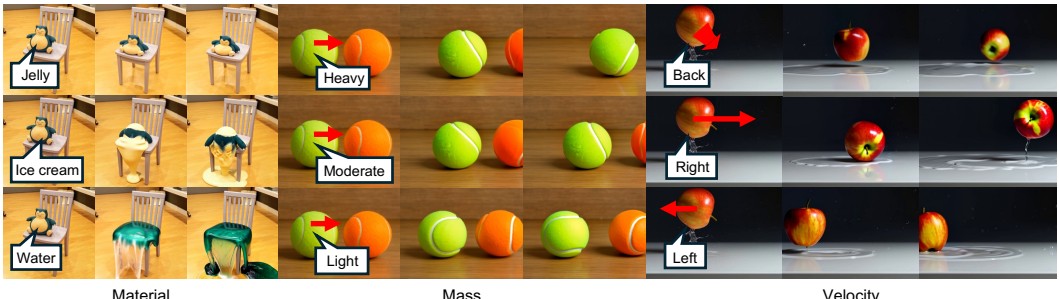

Figure 5: Qualitative results under different physical property conditions. Varying the *material* (e.g., jelly, ice cream, water) changes the deformation behavior, altering the *mass* (heavy, moderate, light) affects motion dynamics, and modifying the *velocity* (backward, rightward, leftward) controls directional trajectories.

**Physical Dynamics.** We demonstrate in Fig. 5 that our framework allows flexible control over different input physical parameters (material, mass, and velocity), and generates results that faithfully reflect the specified properties. Since we use video priors to synthesize faithful videos from simulation results, this enables handling out-of-domain physics scenarios (e.g., Snorlax with ice cream or water-like properties) where data-driven generation methods break down.

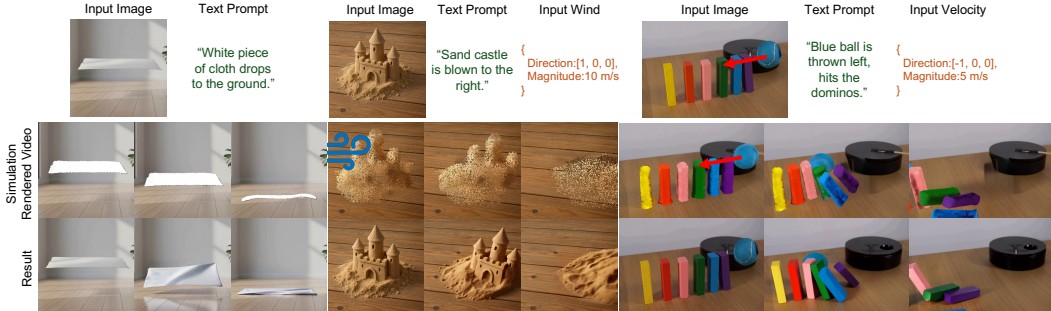

Figure 6: Qualitative results on complex, non-local physical phenomena. Left: a white piece of cloth falling and deforming on the ground. Middle: wind causing sand dispersion in our sand-castle scene. Right: high-speed ball impact producing long-range domino cascades through object compositing on two scenes from the Physics-IQ benchmark.

**Complex and Non-Local Physical Phenomena.** To further assess the scalability of 3DPHYSVIDEO beyond localized interactions, we evaluate our framework on large-scale, non-local physical scenarios, as illustrated in Fig. 6. First, we additionally simulate a free-falling white piece of cloth by adjusting material parameters such as Young's modulus to approximate real fabrics. This enables our method to reproduce basic cloth behaviors, including folding and deformation during impact. Second, we simulate a wind-driven scene where strong lateral airflow erodes and disperses a sand castle across the scene. Lastly, we construct a multi-object collision setup in which a high-speed ball impacts a dense array of dominoes, producing long-range cascaded interactions. In all

cases, our method preserves the input appearance while generating coherent, globally coupled motion faithful to the simulated dynamics, demonstrating robustness in handling complex non-local physical effects.

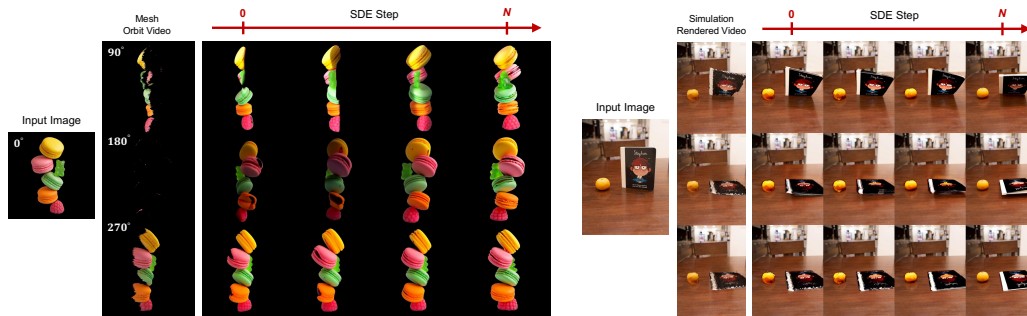

Figure 7: Ablation on the number of Consistency-Guided Flow SDE steps. Different camera orbit angles (90°, 180°, 270°) are shown for the mesh orbit video. The second column (SDE step = 0) shows the result of *standard generation process* without SDE optimization, while the last column (SDE step = $N$) presents our final result.

**Effects of SDE Optimization.** Fig. 7 shows the effectiveness of our Consistency-Guided Flow SDE by varying the number of SDE optimization steps in both stages of our pipeline. With zero refinement steps, unseen regions in orbit views (Stage 1) remain incomplete while raw simulation frames (Stage 2) appear unrealistic with distorted textures. As the number of refinement steps increases, the orbit views progressively fill in missing areas producing geometrically consistent reconstructions, while the simulation frames become increasingly photorealistic with stable textures faithful to the input image.

**Applications.** Our Consistency-Guided Flow SDE $\Phi_{CF}$ is broadly applicable across diverse domains. First, while our main task achieves fine-level simulated motion to video generation via a masked inversion strategy, $\Phi_{CF}$ also supports coarse-level, general motion-controllable video generation without the masked inversion. This enables any off-the-shelf video model to operate like Go-with-the-Flow (Burgert et al., 2025), even training-free. Furthermore, $\Phi_{CF}$ can exploit non-visual inductive biases, such as text alignment, instead of the visual-consistency bias with the input image in our task. This supports applications such as text-guided video editing and refinement of alignment with detailed text prompts. Examples and implementation details can be found in the supplementary material Sec. I.

## 5 CONCLUSIONS

We introduced a novel training-free pipeline for generating a physically plausible video from a single input image. We believe this work represents a significant step forward by demonstrating how off-the-shelf video generation models can be repurposed without training as novel-view synthesizers and simulator-guided renderers for single-view 3D reconstruction and photorealistic video generation. The core of our framework, the proposed Consistency-Guided Flow SDE, while the lack of consistency in this work reveals the model's inherent consistency bias, can serve as a general-purpose bias enforcer for other applications requiring different inductive biases, such as text alignment. Future work will explore these broader applications and extend to more complex physics dynamics for real-world scenario modeling.

**Limitations.** Our approach faces limitations when handling volumetric liquids, such as deep water bodies, because our surface-based reconstruction cannot capture occluded geometries like the bottom of a liquid volume. Additionally, reliance on VLMs to estimate physical parameters, including Young's modulus and density, can introduce errors that propagate to the final simulation. Visual examples of these failure cases are detailed in Sec. H of the supplementary material. Future work will focus on addressing volumetric liquid reconstruction and enhancing the robustness of physical property inference.

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

## A   DETAILED DERIVATION FOR CONSISTENCY-GUIDED FLOW SDE ITERATIVE FORMULA

To derive Eq. 5, we start from our objective in Eq. 4:

$$p^* = \arg\max_p \ \mathbb{E}_{\mathbf{z}_\tau \sim p} \left[ C(\mathbf{z}_\tau, \mathbf{z}_I) \right] - \frac{1}{\beta} \mathbb{D}_{\mathrm{KL}}(p \,\|\, q). \tag{6}$$

By expanding the KL divergence term and rewriting the objective in its minimizing form, we can rewrite it as:

$$\arg\min_p \mathbb{E}_{\mathbf{z}_\tau \sim p} \left[ \frac{1}{\beta} \log \frac{p(\mathbf{z}_\tau)}{q(\mathbf{z}_\tau)} - C(\mathbf{z}_\tau, \mathbf{z}_I) \right] = \arg\min_p \mathbb{E}_{\mathbf{z}_\tau \sim p} \left[ \log \frac{p(\mathbf{z}_\tau)}{q(\mathbf{z}_\tau)} - \beta C(\mathbf{z}_\tau, \mathbf{z}_I) \right]. \tag{7}$$

Following the exponential tilting result for KL-constrained objectives in (Rafailov et al., 2023)), the optimal distribution is:

$$p^*(\mathbf{z}_\tau) = \frac{q(\mathbf{z}_\tau) \, \exp\!\big(\beta C(\mathbf{z}_\tau, \mathbf{z}_I)\big)}{\int q(\mathbf{z}'_\tau) \, \exp\!\big(\beta C(\mathbf{z}'_\tau, \mathbf{z}_I)\big) \, d\mathbf{z}'_\tau}. \tag{8}$$

Taking the score of this optimal distribution:

$$\nabla_{\mathbf{z}_\tau} \log p^*(\mathbf{z}_\tau) = \beta \, \nabla_{\mathbf{z}_\tau} C(\mathbf{z}_\tau, \mathbf{z}_I) + \nabla_{\mathbf{z}_\tau} \log q(\mathbf{z}_\tau). \tag{9}$$

This score function directly gives us the drift term for the overdamped Langevin SDE (Risken, 1989; Song et al., 2021):

$$d\mathbf{z}_k = \left[ \beta \, \nabla_{\mathbf{z}_\tau} C(\mathbf{z}_\tau, \mathbf{z}_I) + \nabla_{\mathbf{z}_\tau} \log q(\mathbf{z}_\tau) \right] dk + \sqrt{2} \, d\mathbf{W}_k. \tag{10}$$

where $\mathbf{W}_k$ is a standard Wiener process. To implement this, we apply our decomposed velocity models in Eq. 3. We approximate the $\nabla_{\mathbf{z}_\tau} C(\mathbf{z}_\tau, \mathbf{z}_I)$ using our consistency bias model $v_c = v_\theta - v_\epsilon$ and approximate the score function using the denoising bias model $v_\epsilon$ based on the flow ODE structure:

$$\nabla_{\mathbf{z}_\tau} C(\mathbf{z}_\tau, \mathbf{z}_I) \approx v_c = v_\theta - v_\epsilon, \quad \nabla_{\mathbf{z}_\tau} \log q(\mathbf{z}_\tau) \approx -\frac{\mathbf{z}_\tau - (1 - \tau)(\mathbf{z}_\tau + \tau v_\epsilon)}{\tau^2}. \tag{11}$$

By substituting these approximations into Eq. 10:

$$d\mathbf{z}_k = \left[ \beta \, (v_\theta - v_\epsilon) - \frac{\mathbf{z}_\tau - (1 - \tau)(\mathbf{z}_\tau + \tau v_\epsilon)}{\tau^2} \right] dk + \sqrt{2} \, d\mathbf{W}_k. \tag{12}$$

Applying the Euler-Maruyama discretization (Kloeden & Pearson, 1977; Gianfelici, 2008). with step size $\gamma$ gives us the update rule:

$$\begin{aligned} \mathbf{z}_\tau^{(n+1)} &= \mathbf{z}_\tau^{(n)} + \left[ \beta \, (v_\theta - v_\epsilon) - \frac{\mathbf{z}_\tau^{(n)} - (1 - \tau)(\mathbf{z}_\tau^{(n)} + \tau v_\epsilon)}{\tau^2} \right] \gamma + \sqrt{2\gamma} \, \boldsymbol{\epsilon}^{(n)} \\ &= (1 - \frac{\gamma}{\tau})\mathbf{z}_\tau^{(n)} + \beta\gamma \, v_\theta - (\beta - \frac{1 - \tau}{\tau})\gamma v_\epsilon + \sqrt{2\gamma} \, \boldsymbol{\epsilon}^{(n)}, \quad \boldsymbol{\epsilon}^{(n)} \sim \mathcal{N}(0, \mathbf{I}). \end{aligned} \tag{13}$$

This discretized iterative formula ensures convergence to the optimal distribution $p^*(\mathbf{z}_\tau)$ that satisfies our objective in Eq. 4, as established by the exponential tilting result and the convergence properties of Euler-Maruyama method. Notably, when we choose $\beta = \frac{1-\tau}{\tau}$, the coefficient of $v_\epsilon$ vanishes: $(\beta - \frac{1-\tau}{\tau})\gamma = 0$, causing the $v_\epsilon$ terms to completely cancel out. This yields a simplified final iterative formula in Eq. 5 that depends only on the known velocity model $v_\theta$, eliminating the need for explicit knowledge of the denoising bias model $v_\epsilon$ while maintaining convergence to the desired optimal distribution.

---

**Algorithm 1** Consistency-Guided Flow SDE $\Phi_{\mathrm{CF}}$

---

**Require:** Input image $\mathbf{I}$, Input rendered video $\{\mathbf{f}_i\}$, Video mask $\mathbf{M}$, Background image $\mathbf{I}^{\mathrm{bg}}$, SDE target timestep $\tau$, SDE optimization iterations $N$, One-step flow model generation process $F(\cdot, v_\theta, t)$, One-step flow model inversion process $F^{-1}(\cdot, v_\theta, t)$, Encoding process $E(\cdot)$, Decoding process $D(\cdot)$

1: $(\mathbf{z}, \mathbf{z}_I, \mathbf{z}^{\mathrm{bg}}) \leftarrow E(\{\mathbf{f}_i\}, \mathbf{I}, \mathbf{I}^{\mathrm{bg}})$             $\triangleright$ encode each input to its corresponding latent

2: $\mathbf{z}_0^{\mathrm{inv}} \leftarrow \mathbf{z}$

3: **for** $t = 0 \ldots \tau - 1$ **do**

4:      $\mathbf{z}_{t+1}^{\mathrm{inv}} \leftarrow F^{-1}(\mathbf{z}_t^{\mathrm{inv}}, v_\theta(\mathbf{z}_t^{\mathrm{inv}}, \mathbf{z}_I, t), t)$

5: **end for**

6: $\boldsymbol{\epsilon} \sim \mathcal{N}(0, \mathbf{I})$

7: $\mathbf{z}_\tau^{\mathrm{noisy}} \leftarrow \begin{cases} (1 - \tau) \cdot \mathbf{z} + \tau \cdot \boldsymbol{\epsilon} & \text{(stage 1)} \\ (1 - \tau) \cdot \mathbf{z}^{\mathrm{bg}} + \tau \cdot \boldsymbol{\epsilon} & \text{(stage 2)} \end{cases}$

8: $\mathbf{z}_\tau \leftarrow \mathbf{M} \cdot \mathbf{z}_\tau^{\mathrm{inv}} + (1 - \mathbf{M}) \cdot \mathbf{z}_\tau^{\mathrm{noisy}}$

9: **for** $n = 0 \ldots N - 1$ **do**

10:      $\boldsymbol{\epsilon}^{(n)} \sim \mathcal{N}(0, \mathbf{I})$

11:      $\hat{\mathbf{z}}_\tau^{(n+1)} \leftarrow (1 - \frac{\gamma}{\tau})\mathbf{z}_\tau^{(n)} + \frac{1-\tau}{\tau}\gamma v_\theta(\mathbf{z}_\tau^{(n)}, \mathbf{z}_I, \tau) + \sqrt{2\gamma}\, \boldsymbol{\epsilon}^{(n)}$         $\triangleright$ Eq. 5

12:      $\mathbf{z}_\tau^{(n+1)} \leftarrow \begin{cases} \mathbf{M} \cdot \mathbf{z}_\tau^{(n)} + (1 - \mathbf{M}) \cdot \hat{\mathbf{z}}_\tau^{(n+1)} & \text{(stage 1)} \\ \mathbf{M} \cdot \hat{\mathbf{z}}_\tau^{(n+1)} + (1 - \mathbf{M}) \cdot \mathbf{z}_\tau^{(n)} & \text{(stage 2)} \end{cases}$

13: **end for** $\mathbf{z}_\tau^* \leftarrow \mathbf{z}_\tau^{(N)}$

14: **for** $t = \tau \ldots 1$ **do**

15:      $\mathbf{z}_{t-1}^* \leftarrow F(\mathbf{z}_t^*, v_\theta(\mathbf{z}_t^*, \mathbf{z}_I, t), t)$

16: **end for** $\mathbf{z}^* \leftarrow \mathbf{z}_0^*$

17: $\mathbf{V} \leftarrow D(\mathbf{z}^*)$             $\triangleright$ decode latent to pixel video

18: **Output:** Final video $\mathbf{V}$

---

# B    Algorithm Implementation and Analysis

We provide the practical implementation of our Consistency-Guided Flow SDE $\Phi_{\mathrm{CF}}$. The Alg. 1 takes $\{\mathbf{f}_i^{\mathrm{orb}}\}_{i=1}^K$ or $\{\mathbf{f}_i^{\mathrm{sim}}\}_{i=1}^L$ as the input rendered video $\{\mathbf{f}_i\}$ and $\mathbf{M}^{\mathrm{orb}}$ or $\mathbf{M}^{\mathrm{sim}}$ as the video mask $\mathbf{M}$, and outputs $\mathbf{V}^{\mathrm{orb}}$ or $\mathbf{V}^{\mathrm{sim}}$ as the Final video $\mathbf{V}$. Note that only two lines (7, 12) differ between the two stages.

To clarify the algorithm's unified approach, we provide detailed explanations for both use cases, highlighting the minimal but crucial differences between the two stages:

**For Stage 1.** The rendered mesh orbit video $\{\mathbf{f}_i^{\mathrm{orb}}\}_{i=1}^K$ shows accurate localization in the 3D world coordinate system for the regions visible in the input image. Therefore, given a mask $\mathbf{M}$ for this region, our aim is to preserve and utilize it as geometric guidance to fill the remaining empty regions so that they align semantically with the input image $I$. To achieve this, in line 12, we continuously inject $\mathbf{z}_\tau^{\mathrm{inv}}$ corresponding to the mask $\mathbf{M}$ region from $\mathbf{z}_\tau$ at every optimization iteration, while updating only the unmasked part. This strategy ensures that during optimization, the geometry guidance from the input rendered video $\{\mathbf{f}_i^{\mathrm{orb}}\}_{i=1}^K$ is accurately followed while enabling the filling of empty regions to obtain the optimal latent $\mathbf{z}_\tau^*$.

**For Stage 2.** The simulation-rendered video $\{\mathbf{f}_i^{\mathrm{sim}}\}_{i=1}^L$ provides accurate motion dynamics but exhibits unrealistic visual appearance. Therefore, the objective of Stage 2 is to covert it into photorealistic video that is visually coherent with $\mathbf{I}$. To achieve this, we only need simple modifications at two lines of the Stage 1 algorithm. First, in line 7, we use $\mathbf{z}^{\mathrm{bg}}$ instead of $\mathbf{z}$. This prevents the simulated motion from unintentionally spreading into regions where it is not required. Second, in line 12, we only update the masked region $\mathbf{M}$ (opposite to Stage 1) to optimize the appearance of the inverted simulation part. As a result, the regions with motion are optimized while their area is controlled by the mask $\mathbf{M}$, and the unmasked static regions are injected with noise only once at initialization. This enables the $\mathbf{M}$ region to be refined in appearance while following the simulated motion guidance well, and allows the remaining regions to have natural backgrounds (shadows, lighting, and motion) thanks to the initial noise $\boldsymbol{\epsilon}$, resulting in a physics-faithful and photorealistic final video $\mathbf{V}^{\mathrm{sim}}$.

## C    POINT CLOUD AND MESH PROCESSING DETAILS

For completeness, we describe the detailed procedure for obtaining simulation-ready meshes and point clouds from the reconstructed orbit video.

**Step 1: Orbit video to point cloud.**    The completed orbit video is passed to VGGT (Wang et al., 2025b), using approximately 10 frames as input, to obtain an initial multi-view point cloud $\mathcal{P}_0$.

**Step 2: Density-based outlier removal.**    To filter low-density outliers, we compute a density measure for each point $p \in \mathcal{P}_0$ as

$$\rho(p) = \frac{1}{d_k(p)^2},$$

where $d_k(p)$ is the distance to the $k$-th nearest neighbor ($k = 10$). Points with density below a fixed threshold $\tau$ are removed.

**Step 3: Ground plane processing.**    The ground plane $\pi$ is detected via RANSAC (Fischler & Bolles, 1981). For each object point cloud, the minimum distance to $\pi$ is computed; if sufficiently small, the object is considered to be in contact with the ground. In this case, points are projected onto $\pi$ and merged into the cloud, while points located below $\pi$ are pruned.

**Step 4: Poisson mesh reconstruction.**    The filtered point cloud is converted into a watertight surface mesh using Poisson surface reconstruction (Kazhdan et al., 2006) with depth parameter $d = 6$.

**Step 5: Volumetric sampling.**    Finally, volumetric sampling is performed within each reconstructed mesh to obtain a simulation-ready particle cloud. We generate between 20k and 50k particles per object, depending on its size, to balance accuracy and efficiency.

This pipeline yields clean, watertight, and volumetrically sampled object representations suitable for subsequent MPM simulation.

## D    MATERIAL POINT METHOD (MPM) FORMULATION

For completeness, we briefly describe the Material Point Method (MPM) adopted in our simulation pipeline. MPM bridges Lagrangian particles with an Eulerian background grid: particles carry material properties, while force computations and collisions are handled on the grid.

**Particle state.**    Each particle $p$ is associated with mass $m_p$, position $\mathbf{x}_p$, velocity $\mathbf{v}_p$, and deformation gradient $\mathbf{F}_p$. A background grid with nodes $i$ accumulates quantities from nearby particles via interpolation weights $w_{ip}$.

**Particle-to-Grid (P2G).**    Mass and momentum are transferred from particles to grid nodes:

$$m_i = \sum_p w_{ip} m_p, \tag{14}$$

$$\mathbf{v}_i = \frac{1}{m_i} \sum_p w_{ip} m_p \mathbf{v}_p. \tag{15}$$

**Grid update.**    The stress on each particle is computed from a constitutive potential $\Psi(\mathbf{F}_p)$. The first Piola–Kirchhoff stress is

$$\mathbf{P}_p = \frac{\partial \Psi}{\partial \mathbf{F}_p}. \tag{16}$$

The force at grid node $i$ is

$$\mathbf{f}_i = -\sum_p V_p \mathbf{P}_p \nabla w_{ip}, \tag{17}$$

where $V_p$ is the particle volume. Grid velocities are updated as

$$\mathbf{v}_i \leftarrow \mathbf{v}_i + \Delta t\, \frac{\mathbf{f}_i}{m_i} + \Delta t\, \mathbf{g}, \tag{18}$$

with timestep $\Delta t$ and gravity $\mathbf{g}$.

**Grid-to-Particle (G2P).** The updated grid state is interpolated back to the particles:

$$\mathbf{v}_p \leftarrow \sum_i w_{ip}\mathbf{v}_i, \tag{19}$$

$$\mathbf{x}_p \leftarrow \mathbf{x}_p + \Delta t\, \mathbf{v}_p. \tag{20}$$

The deformation gradient is evolved as

$$\mathbf{F}_p \leftarrow (\mathbf{I} + \Delta t\, \nabla\mathbf{v}_p)\, \mathbf{F}_p, \qquad \nabla\mathbf{v}_p = \sum_i \mathbf{v}_i \nabla w_{ip}. \tag{21}$$

This hybrid particle–grid update allows simulating diverse materials, including elastic solids, fluids, and granular matter, depending on the choice of constitutive model $\Psi$. The MPM framework naturally handles large deformations, material mixing, and collisions, making it suitable for our simulation-to-video setting.

# E  EVALUATION PROTOCOL

## E.1  HUMAN EVALUATION PROTOCOL

To complement the automatic metrics, we conducted a human evaluation following the protocol introduced in PhysGen3D. A total of 20 participants were recruited to assess the quality of generated videos. The evaluation was performed on 10 representative scenes, each rendered by 7 different methods, resulting in 70 videos in total. For each video, participants were asked to answer three questions corresponding to three quality dimensions: physical realism, photorealism, and semantic consistency.

At the beginning of the study, participants were provided with the following instruction:

> *We want to evaluate the quality of the generated video. You will be asked to assess it from the three perspectives: physical realism, photorealism, and semantic consistency.*

The three evaluation criteria were described in detail as follows:

- **Physical Realism** measures how realistically the video follows physical rules. Participants were asked to consider whether the video represents physical properties such as elasticity and friction, and whether the movements and interactions of objects behave plausibly and consistently with real-world expectations.

- **Photorealism** evaluates the visual fidelity of the video, including whether there are visual artifacts or discontinuities, and whether lighting, shadow, texture, and material details closely resemble real-world appearances.

- **Semantic Consistency** examines how well the generated video aligns with the provided input text and reference image.

Each video was rated by answering the following three questions: 1) *The video is physically realistic.* 2) *The video is photorealistic.* 3) *The video is consistent with the input image and input text "{text prompt}".*

Responses were collected on a 5-point Likert scale (1 = strongly disagree, 5 = strongly agree). The order of videos was randomized across participants to mitigate ordering bias.

### E.2 GPT-BASED EVALUATION PROTOCOL

For completeness, we include the details of the GPT-based evaluation protocol, which we directly adopt from PhysGen3D (Chen et al., 2025a). In this protocol, GPT is prompted to assess generated videos along three axes: *Physical Realism*, *Photorealism*, and *Semantic Consistency*. The evaluation is performed on evenly sampled frames from each video, together with the original input image and the motion instructions.

Specifically, GPT is instructed with the following template:

> I would like you to evaluate the quality of {num_videos} generated videos based on the following criteria: physical realism, photorealism, and semantic consistency.
>
> The evaluation will be based on {num_frames} evenly sampled frames from each video. Given the original image and the following instructions: '{instructions}', please evaluate the quality of each video on the three criteria mentioned above.
>
> Note that:
>
> Physical Realism measures how realistically the video follows the physical rules and whether the video represents real physical properties like elasticity and friction. To discourage completely stable video generation, we instruct respondents to penalize such cases.
>
> Photorealism assesses the overall visual quality of the video, including the presence of visual artifacts, discontinuities, and how accurately the video replicates details of light, shadow, texture, and materials.
>
> Semantic Consistency evaluates how well the content of the generated video aligns with the intended meaning of the text prompt.
>
> Please provide the following details for each video, scores should be ranging from 0–1, with 1 to be the best: {score_lines}
>
> Note that your output should strictly follow the above format, with a ';' after each score. Do not give any other explanations.

We emphasize that this protocol and prompt design are not introduced by us, but are directly inherited from PhysGen3D for completeness. Our goal is to ensure consistency with prior work, rather than proposing a new evaluation methodology.

## F ANALYSIS ON SDE HYPERPARAMETERS

**Regularization Parameter $\beta$.** The regularization parameter $\beta$ balances the term and the KL divergence term, so it is not necessary to set $\beta = \frac{1-\tau}{\tau}$ to entirely eliminate the $v_\epsilon$ term. In Eq. 4, when $\beta \to \infty$, the KL divergence term is ignored, and the optimal distribution $p^*$ will not be able to maintain the original distribution $q = N((1-\tau)\boldsymbol{\mu}, \tau^2 \mathbf{I})$. When $\beta \to 0$, the consistency term is ignored, so the original distribution is well preserved but consistency optimization does not occur. Therefore, a non-extreme choice of $\beta$ is required, and among these, our choice of $\beta = \frac{1-\tau}{\tau}$ is empirically the most practical. We verified this through additional experiments. For experiments with different $\beta$ values in the SDE, we explicitly needed a additional denoising bias model $v_\epsilon$ in Eq. 13. To achieve this, we modified the existing $v_\theta(\mathbf{z}_t, \mathbf{z}_I, t)$ to remove the consistency bias by conditioning it on the first frame of the video latent $\mathbf{z}_t$ instead of the image latent $\mathbf{z}_I$ at every SDE optimization step, i.e., $v_\epsilon = v_\theta(\mathbf{z}_t, \mathbf{z}_t[0], t)$. As show in Fig. 8, when $\beta \to \infty$, the latent cannot maintain the original distribution, resulting in distorted videos. When $\beta \to 0$, consistency is not optimized, producing non-photorealistic videos. In contrast, within an appropriate range of $\beta$, the desired videos are generated, showing that our Consistency-Guided Flow SDE works well with low sensitivity to the hyperparameter $\beta$. However, among these $\beta$, only our choice entirely eliminates $v_\epsilon$. The reason is that at every SDE iteration, we only need $v_\theta$ without requiring both $v_\theta$ and $v_\epsilon$, this results in only half the number of function evaluations (NFEs) for optimization, making it the most practical choice.

**Iterations $N$.** The parameter specifies the number of SDE optimization steps applied via Eq. 5 to achieve consistency with the input image. As shown in Fig. 7, increasing iterations progressively

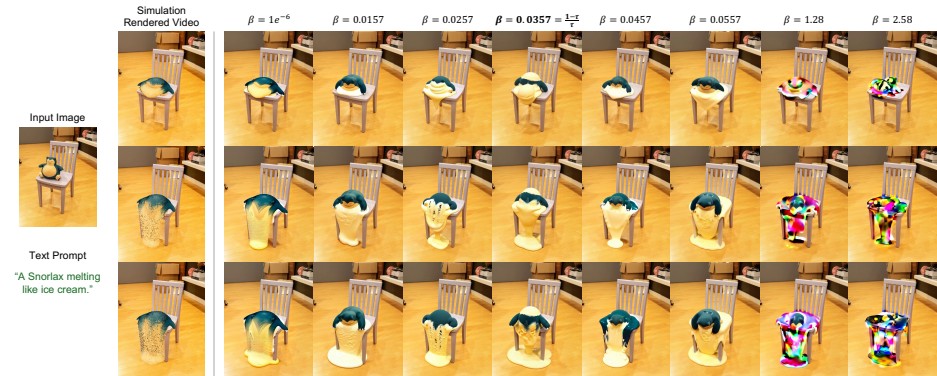

Figure 8: Comparison of generated videos across various $\beta$ values. $\beta = 0.0357 = \frac{1-\tau}{\tau}$ is our choice that entirely eliminates the $v_\epsilon$ term. $\beta = 1e^{-6}$ represents the $\beta \to 0$ case, while $\beta = 1.28$ and $\beta = 2.58$ represent the $\beta \to \infty$ case.

generates videos more consistent with the input image. However, more iterations increase computational cost due to higher NFEs (Number of Function Evaluations) from repeated $v_\theta$ predictions. Empirically, we found that once the video model's prior reaches a satisfactory level of consistency, subsequent iterations maintain similar consistency without significant improvement. Therefore, we practically choose $N = 10$ to balance consistency requirements with computational efficiency, in conjunction with other hyperparameter settings for $\gamma$ and $\tau$.

**Step Size $\gamma$.** To ensure convergence of the Euler-Maruyama discretization (Kloeden & Pearson, 1977; Gianfelici, 2008), the step size $\gamma$ should be sufficiently small. We conducted sensitivity experiments to verify the range of $\gamma$ values that guarantee this convergence property. We evaluated various $\gamma$ values by monitoring whether the distribution of the optimized video latent during SDE optimization stably maintains the original distribution through the KL divergence term in our objective (Eq. 4). Specifically, we tracked the norm of the latent at each SDE step, with results shown in Fig. 9. Our experiments reveal that as SDE steps progress, the original distribution is well-preserved for $\gamma \leq 0.8$, whereas larger values fail to maintain distributional stability.

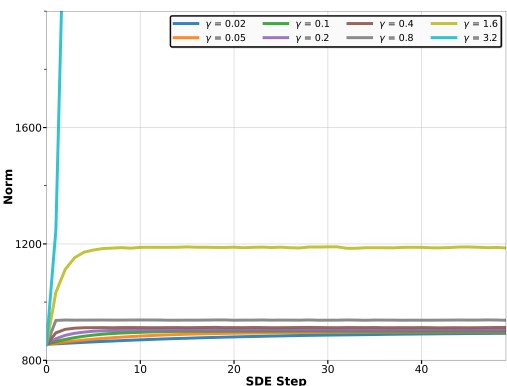

Figure 9: Latent norm at each SDE optimization step for various step sizes $\gamma$.

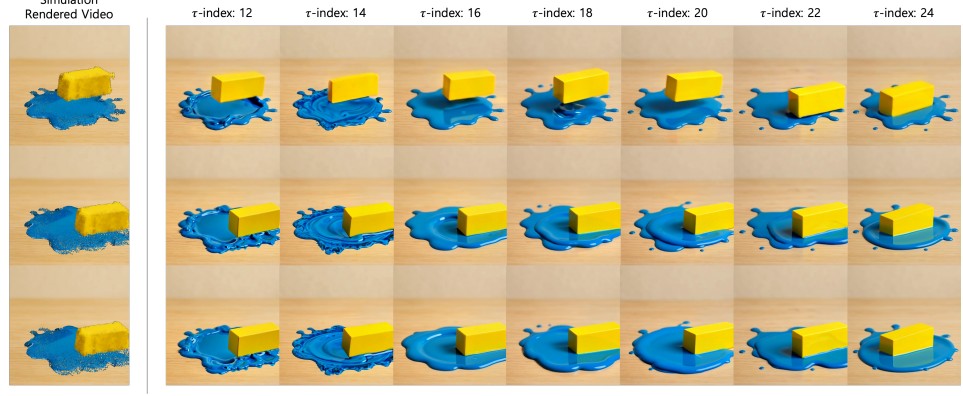

Figure 10: Comparison of generated videos across various target timesteps $\tau$. The $\tau$-index denotes the diffusion step corresponding to $\tau$ (total of 25 steps).

**Target Timestep $\tau$.** In the generation process of diffusion or flow models, early timesteps ($\tau \approx T$) focus on semantic structure (low-frequency components) while later timesteps ($\tau \approx 0$) refine details (high-frequency components). The choice of $\tau$ in our SDE optimization controls which aspects are optimized. Fig. 10 shows results across various $\tau$ values. At $\tau$-index $\approx 25$, the model relies too heavily on its semantic priors and ignores our guidance. At small $\tau$-index, the model cannot generate new components, failing to transform simulation-rendered point clouds into realistic liquids. At $\tau = 20$, the model balances guidance adherence with photorealistic rendering including shadows, liquid reflections, and surface details.

## G  INFERENCE-TIME ANALYSIS

We provide a complete breakdown of the inference time of the full 3DPHYSVIDEO pipeline using a single NVIDIA RTX 3090 GPU. The pipeline consists of two stages: Stage 1 (*Single Image to 3D*) and Stage 2 (*Simulation to Video*), and each stage includes Consistency-Guided Flow SDE ($\Phi_{\text{CF}}$).

**Consistency-Guided Flow SDE.** A video is processed in three latent sections, where each latent section produces 36 frames. Each latent section requires 50 NFEs (20 inversion, 10 SDE refinement, 20 final denoising). The measured cost per NFE is 5.1629 s.

| Component | Value |
|---|---|
| Cost per NFE | 5.1629 s |
| NFEs per latent section | 50 |
| Time per latent section | $50 \times 5.1629 = \mathbf{258.15\,s} \approx \mathbf{4.30\,min}$ |
| Number of latent sections | 3 |
| Time per stage for $\Phi_{\text{CF}}$ | $3 \times 4.30\,\text{min} = \mathbf{12.9\,min}$ |

Table 4: Inference-time breakdown for the Consistency-Guided Flow SDE.

Thus, $\Phi_{\text{CF}}$ requires approximately **12.9 minutes per stage**, corresponding to **25.8 minutes total** across Stage 1 and Stage 2.

**Stage 1 Overhead (Single Image to 3D).** Beyond $\Phi_{\text{CF}}$, Stage 1 performs point-cloud unprojection, mesh reconstruction, volumetric sampling, and segmentation. For the *sand castle* example, these geometry processing steps require **7.3 minutes**.

**Stage 2 Overhead (Simulation to Video).** Prior to applying $\Phi_{\text{CF}}$, Stage 2 performs MPM simulation and point-based rendering. For the *sand castle* example, we simulate 50,000 particles for 200 frames, requiring **7.4 minutes** in total ($\approx 2.2\,\text{s}$ per frame).

**Total Runtime.** Summing all components:

$$\underbrace{12.9}_{\text{Stage 1: } \Phi_{\text{CF}}} + \underbrace{7.3}_{\text{Stage 1: geometry}} + \underbrace{7.4}_{\text{Stage 2: simulation}} + \underbrace{12.9}_{\text{Stage 2: } \Phi_{\text{CF}}} \approx 40\,\text{min}.$$

Thus, generating a full physically plausible and photorealistic video from a single input image requires approximately 40 minutes end-to-end.

## H  FAILURE CASES AND LIMITATIONS

Fig. 11 presents two representative failure modes of our pipeline. The first example (top) shows a yellow block dropped onto the ocean surface. In principle, the object should either float or sink depending on its density. However, our surface-based reconstruction cannot capture the full volumetric geometry of deep water, which prevents accurate modeling of buoyancy-driven interactions. As a result, the block appears to fall directly onto a flat surface rather than interacting with a volumetric liquid. The second example (bottom) illustrates a falling apple. Here, the apple's Young's modulus was incorrectly estimated as $1 \times 10^6$ Pa by the VLM, which is significantly softer than real apples.

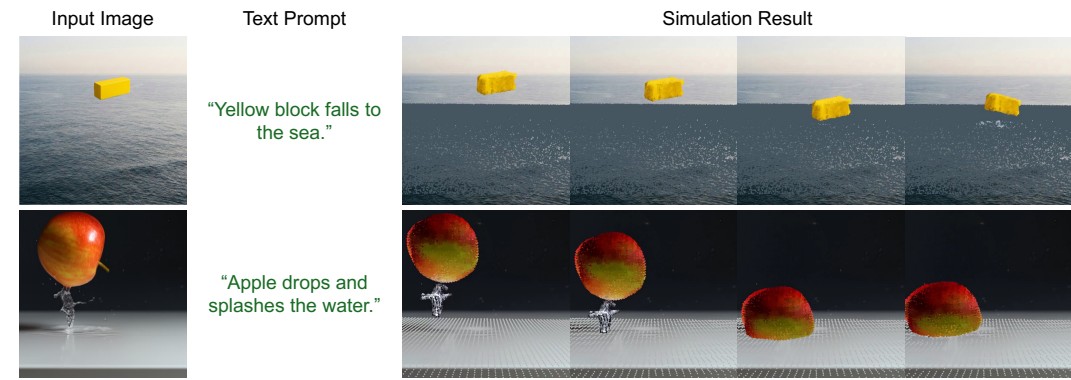

Figure 11: Two representative failure cases from our physics-guided simulation. Top: A yellow block is dropped onto the sea surface, but due to the limitations of our surface-based reconstruction, volumetric water is not faithfully represented, resulting in incorrect floating or sinking behavior. Bottom: An apple with an excessively low Young's modulus ($1 \times 10^6$ Pa) estimated by the VLM deforms unrealistically upon impact. These examples highlight the potential errors in VLM-based physical parameter inference and the need for occasional manual correction.

This leads to unrealistic deformation during impact. Such cases demonstrate that automatic estimation of physical parameters using VLMs may introduce errors that propagate into the simulation, and manual correction is sometimes required to recover physically plausible behavior.

## I    APPLICATIONS OF CONSISTENCY-GUIDED FLOW SDE

**General Motion-Controllable Video Generation.**    The proposed masked inversion process is a method designed to precisely follow fine-grained guidance in our task. Hence, when fine-grained guidance adherence is not necessary, a standard forward process that injects noise can be employed instead of the masked inversion process. This approach enables any off-the-shelf video models to work like Go-with-the-Flow (Burgert et al., 2025) for general motion-controllable video generation, even training-free. As shown in Fig. 12, our Consistency-Guided Flow SDE enables us to achieve realistic dynamic motion that follows the coarse motion prior of the driving video while naturally translating corrupted backgrounds to be consistent with the input image.

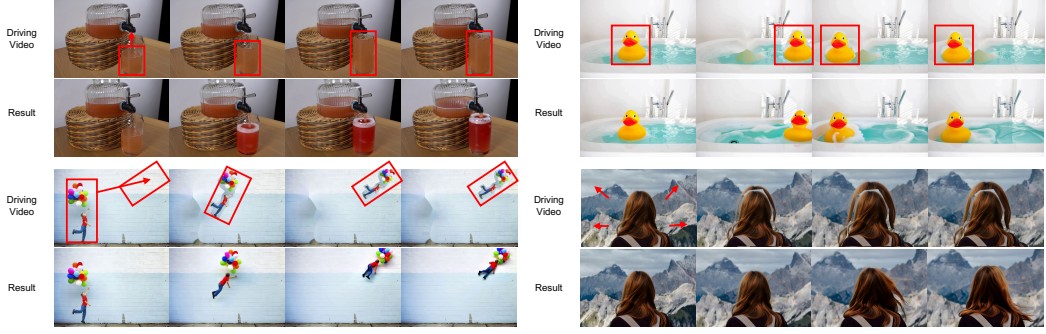

Figure 12: Qualitative results of Consistency-Gudied Flow SDE when given coarse motion. The juice scene (top left) is from the Physics-IQ benchmark, and its driving video is generated following VLIPP (Yang et al., 2025b) using motion planning with VLM-based bounding-box trajectory prediction. The remaining three driving videos come from Go-with-the-Flow (Burgert et al., 2025), created by users specifying regions in the input image and applying Cut and Drag.

**General-Purpose Inductive Bias-Guided Optimization.**    In this work, we create a consistency misalignment situation by providing the I2V flow model with a video inconsistent with the input image, enabling the model's consistency bias to guide our SDE optimization. However, our SDE can be generalized to other inductive biases beyond consistency, such as text alignment. If

we create a situation where the initial video is misaligned with a given text prompt, the model will reveal the text alignment bias $v_{\text{text}}$ that it learned during training. In the same way as $\nabla_{\mathbf{z}_\tau} C(\mathbf{z}_\tau, \mathbf{z}_I) \approx v_c = v_\theta - v_\epsilon$ in Sec. A of the supplementary material, this $v_{\text{text}} = v_\theta - v_\epsilon$ approximates $\nabla_{z_\tau} \text{TextAlignment}(\mathbf{z}_\tau, \mathbf{z}_{\text{text}})$ (where $\mathbf{z}_{\text{text}}$ is the latent of the text feature) and serves as a component of the score term in the SDE for text alignment optimization. Therefore, We can perform text alignment bias-guided SDE optimization without requiring any explicit text alignment metric or input latent changes. Fig. 13 shows our SDE with text alignment inductive bias can enable text-guided video editing using text prompts with different styles from the initial video. Also, while naive inference with the existing model fails to sufficiently reflect detailed text prompts in the generated video, applying SDE optimization progressively refines the video to better align with the detailed text.

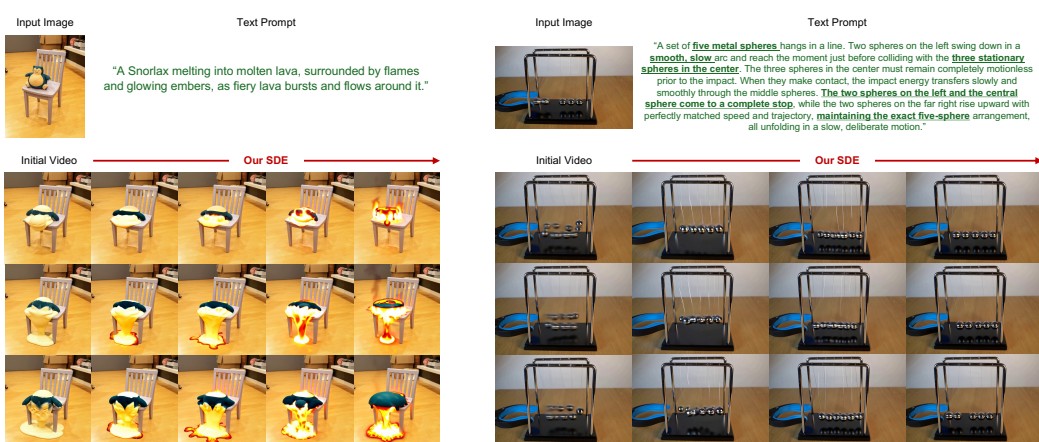

Figure 13: Qualitative results of our SDE with text alignment bias. Left: An initial video from 3DPHYSVIDEO showing a snorlax melting like ice cream is edited to melt into burning lava according to the text prompt. Right: For a detailed text prompt, the initial video generated by FramePack (Zhang & Agrawala, 2025) from that prompt is progressively refined through our SDE to increasingly align with the detailed text prompt (note the **bold** text).

## J  MORE RESULTS FROM 3DPHYSVIDEO

Figure 14: Qualitative comparison with state-of-the-art simulation to video models on the book falling scene.

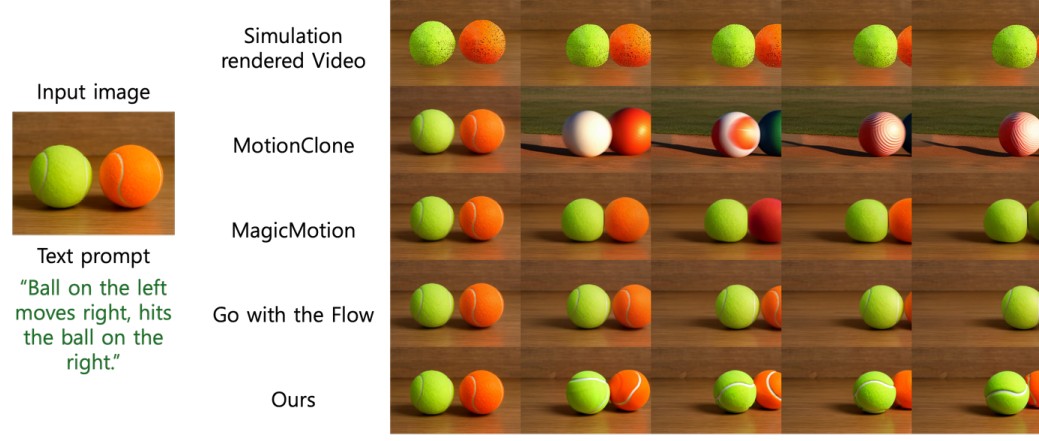

Figure 15: Qualitative comparison with state-of-the-art simulation to video models on the Snorlax deflating scene.

Figure 16: Qualitative comparison with state-of-the-art simulation to video models on the ball collision scene.

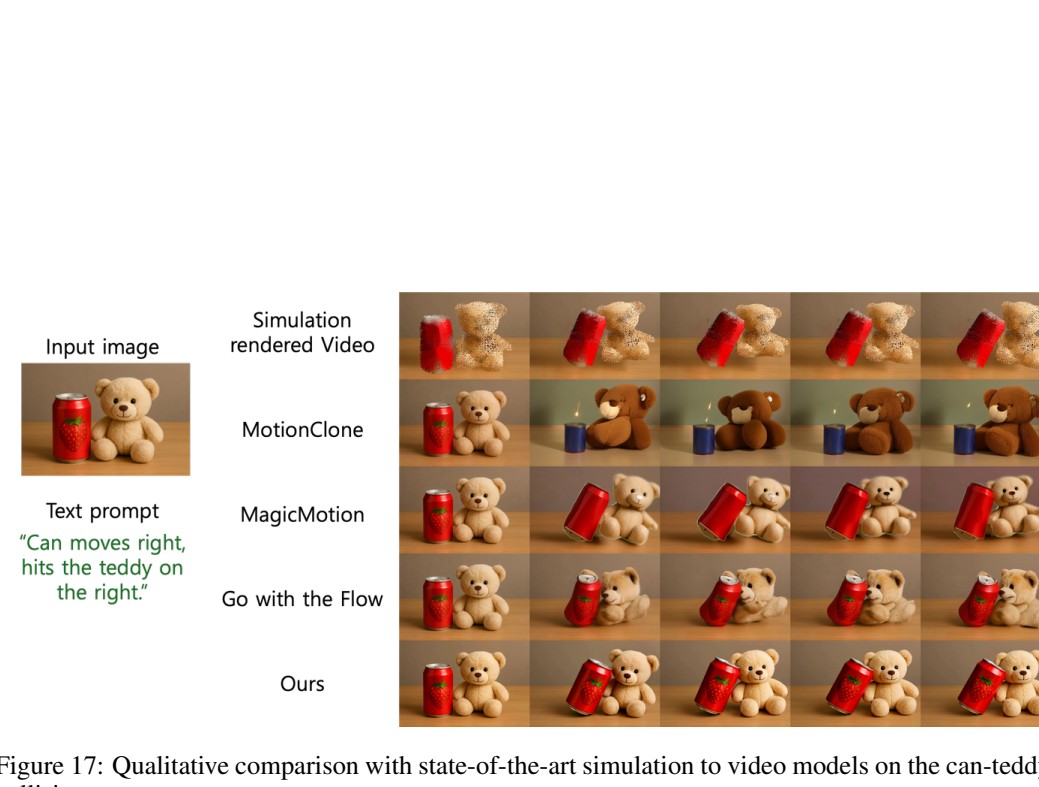

Figure 17: Qualitative comparison with state-of-the-art simulation to video models on the can-teddy collision scene.

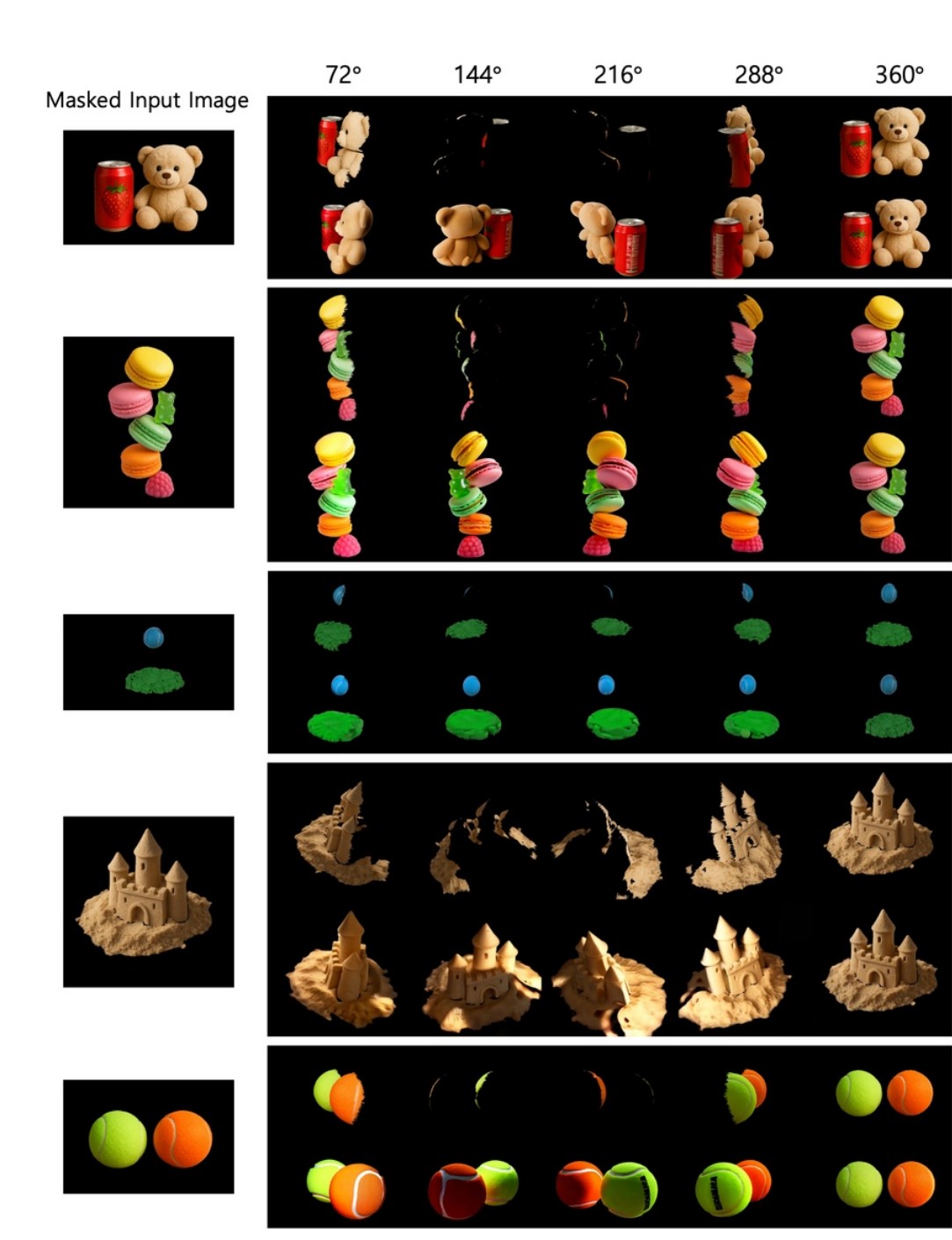

Figure 18: Qualitative results on 360-degree orbit video generation. For five of those examples, the first row is the input mesh orbit video $\{\mathbf{f}_i^{orb}\}$ and the second row is completed orbit video $\mathbf{V}^{orb}$. Our method guides the video model using a rotating mesh, producing visually compelling results with accurate and coherent 3D structure.

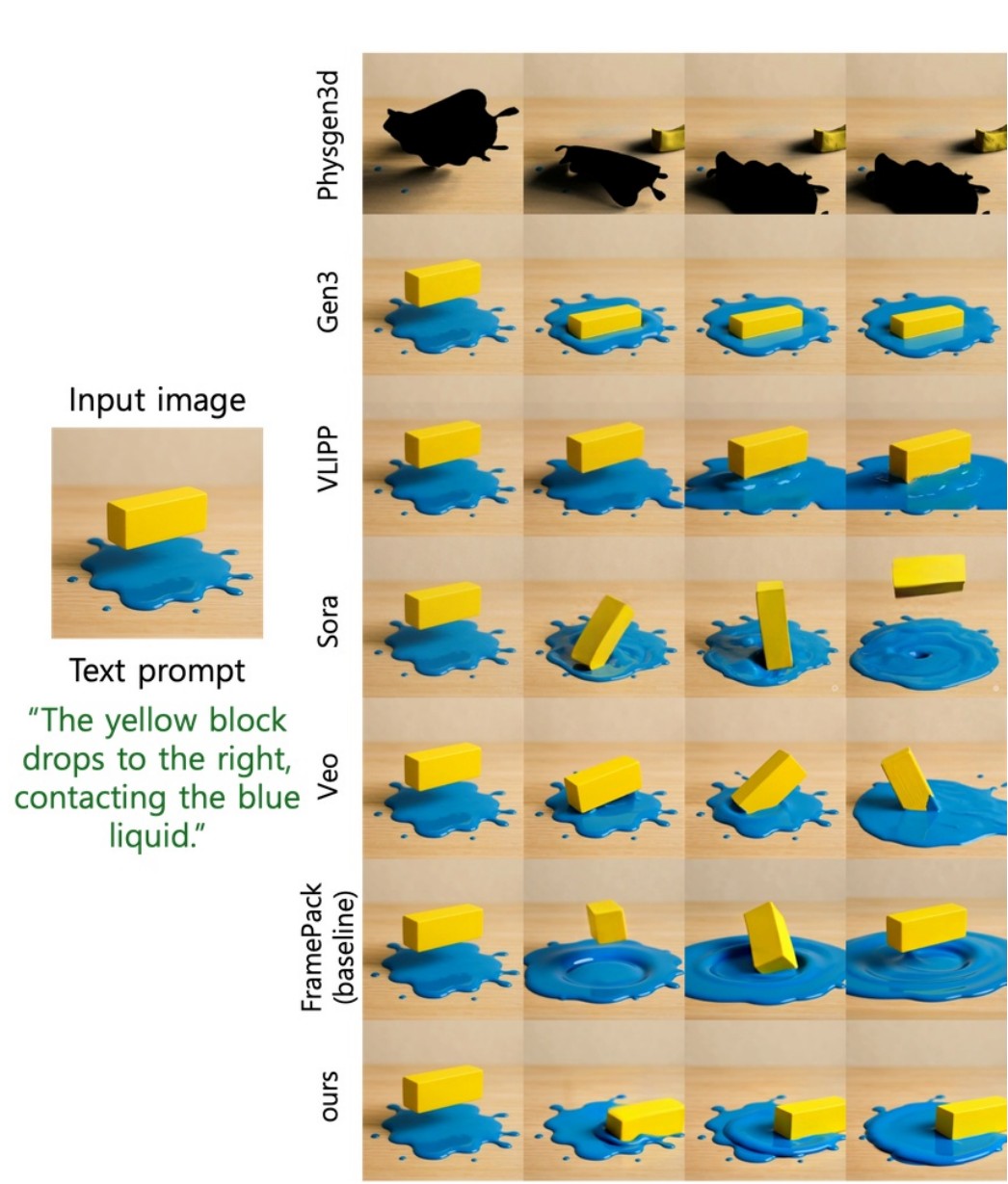

Figure 19: Qualitative comparison on the block falling scene.

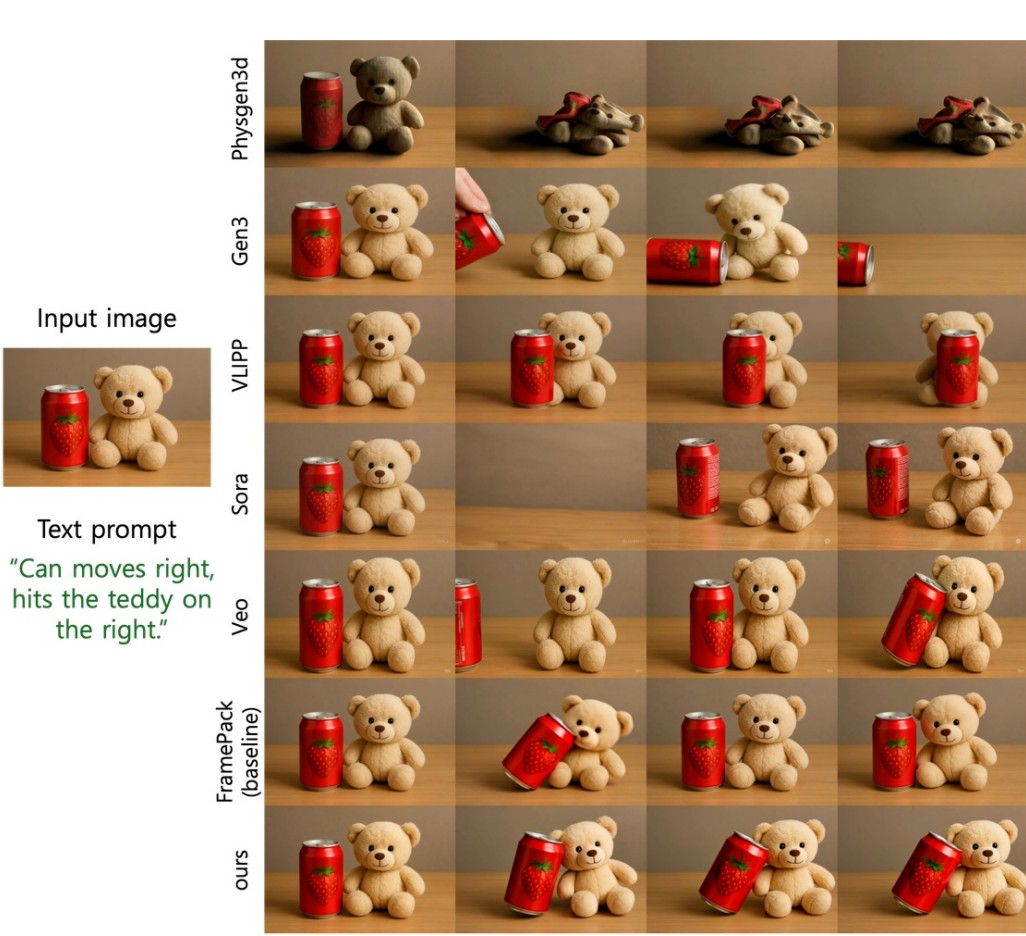

Figure 20: Qualitative comparison on the synthesized can-doll collision scene.

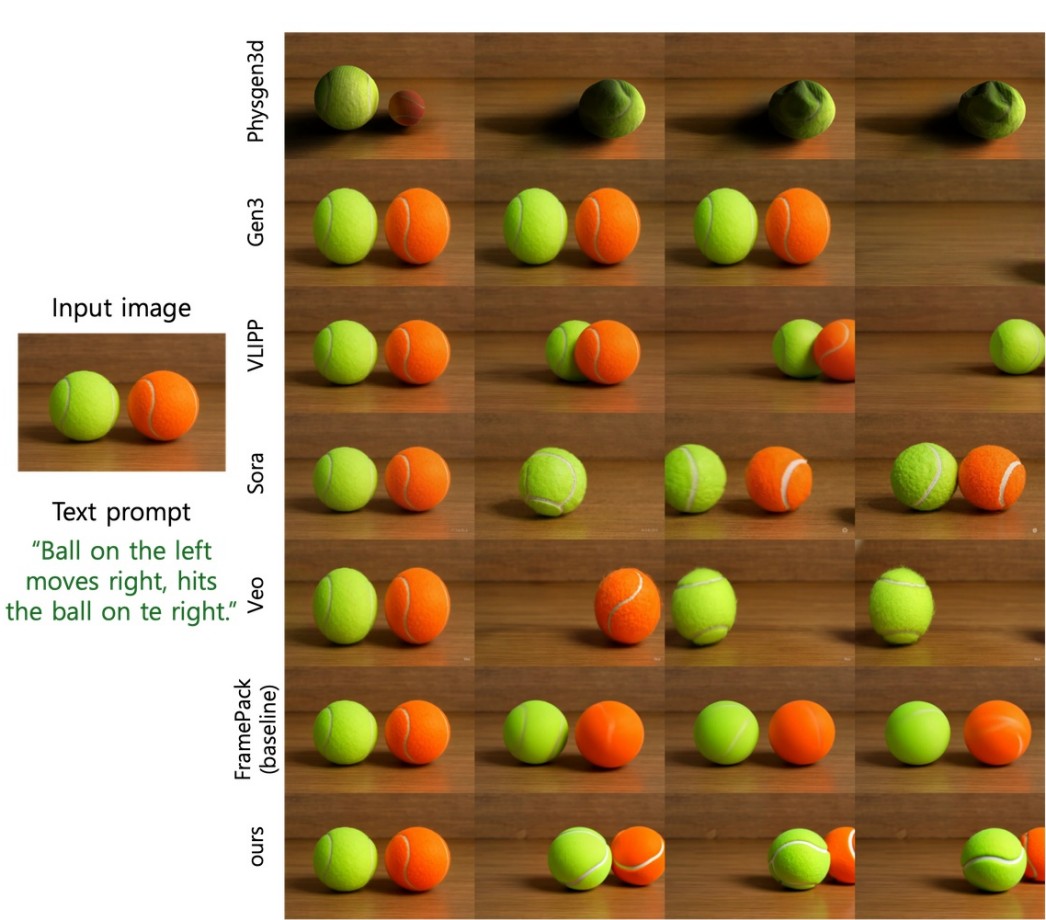

Figure 21: Qualitative comparison on the ball collision scene.

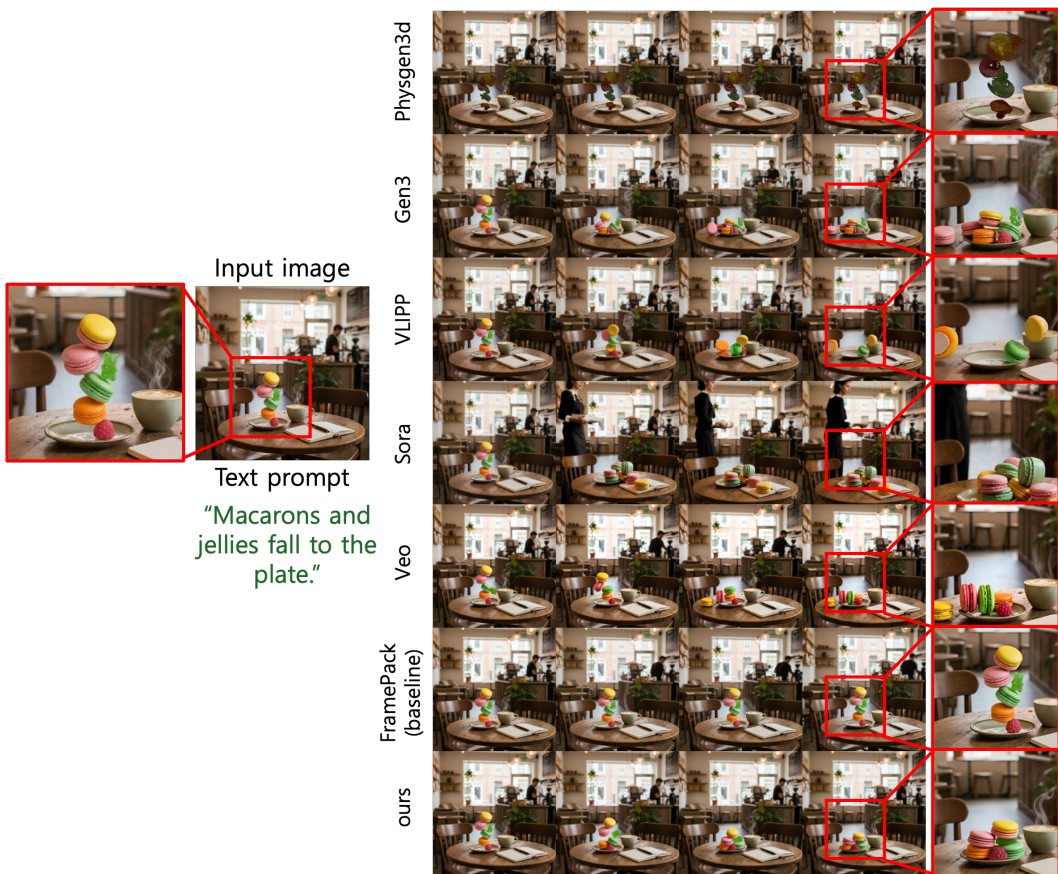

Input image

Text prompt

"Macarons and jellies fall to the plate."

Figure 22: Qualitative comparison. While other models suffer from hallucinating (number of objects vary per frame), ours deliever exact number of objects, since its grounded from physical simulation.

Input image

Text prompt

"The snorlax push toy is deflated and squished."

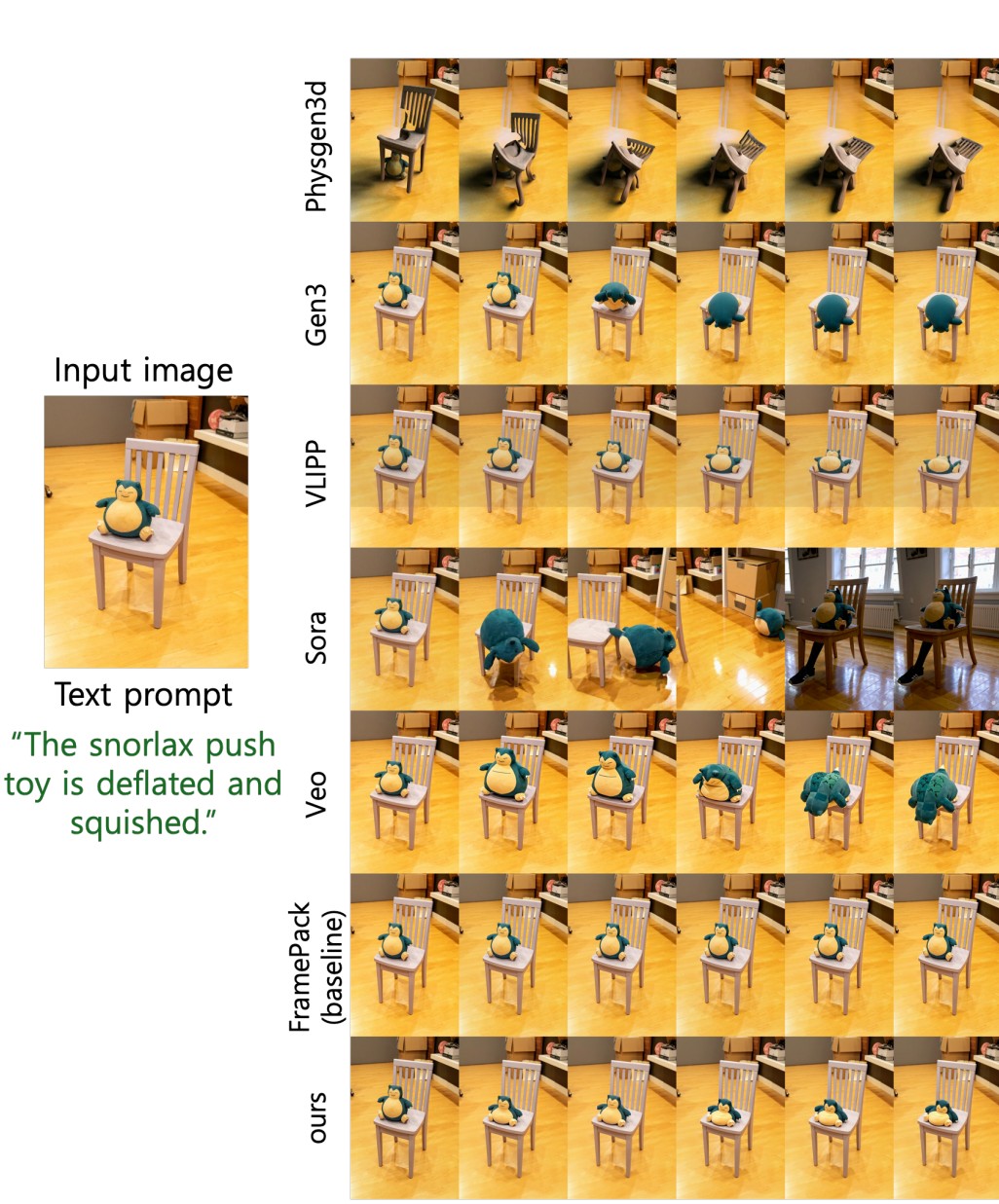

Figure 23: Qualitative comparison on the snorlax deflating scene.

