# OpenReview forum: "3DPhysVideo: 3D Scene Reconstruction and Physical Animation Leveraging a Video Generation Model via Consistency-Guided Flow SDE"
_ICLR.cc/2026/Conference — ICLR 2026 Conference Desk Rejected Submission_

### Official Review · Reviewer_iowy · 2025-10-24

**Soundness:** 3
**Presentation:** 3
**Contribution:** 3
**Rating:** 6
**Confidence:** 3

**Summary:**

This paper proposes a training-free pipeline that generates physically realistic videos from a single image. It repurposes an off-the-shelf image-to-video flow model for two stages: reconstructing full 3D scene geometry using rendered point clouds, and synthesizing final videos guided by Material Point Method physics simulations. The authors also propose Consistency-Guided Flow SDE that decomposes predicted flow velocities to enable effective 3D reconstruction and simulation-guided video generation.

**Strengths:**

- Research on physically realistic video generation is both practical and meaningful.

- The proposed pipeline is well-designed, feasible, and reasonable.

- Experimental results demonstrate performance improvements across multiple scenarios.

**Weaknesses:**

- From the appendix video examples, some cases appear worse than other methods. For instance, in the Apple sample, the back video shows no water splashing when the apple falls. What could be the possible reason for this?

- What is the speed of generating a video sequence, and how does it compare to other methods?

- The paper lacks a discussion of limitations and corresponding analysis.

**Questions:**

Could the authors provide intermediate visual results showing MPM-simulated outputs under different types of interactions, such as solid–fluid collisions, fluid–fluid interactions, and so on?

---

> ### Author Response · Authors · 2025-11-25
> **Rebuttal by Authors**
>
> We appreciate the reviewer’s encouraging and helpful comments, especially the positive feedback on the practicality and significance of pursuing physically realistic video generation, the soundness of our pipeline design, and the consistent performance gains observed across diverse scenarios.
> The following provides our responses to the reviewer feedback, and an additional demo video has been newly included in the supplementary material.
>
> ---
>
> > **W1. No water splash in the backward video of the Apple sample**
> >
>
> In the fluid scenarios of Section 5 in our demo video, only the apple scene has water covering the entire ground. Since we reconstruct the scene by unprojecting point clouds from video, regions farther from the camera inevitably yield lower point density. Therefore, for reliable fluid simulation, we reconstruct liquids only within regions where a sufficient point density can be guaranteed. In the backward simulation of the apple scene, the apple falls outside this reconstructed fluid region, which is why no splashing occurs.
>
> ---
>
> > **W2. Inference Time analysis**
> >
>
> We thank the reviewer for the question regarding the runtime distribution across the stages of our pipeline. Below, we provide a complete inference-time analysis using a single NVIDIA RTX 3090 GPU. The pipeline consists of two major stages : Single Image to 3D (Stage 1) and Simulation to Video (Stage 2). Each stage includes a Consistency-Guided Flow SDE step.
>
> A video is processed in three latent sections, where each latent section generates 36 frames. Each latent section requires 50 NFEs (20 inversion, 10 SDE refinement, and 20 final denoising). The measured cost per NFE is 5.1629 seconds.
>
> **NFE and Latent Section Cost**
>
> | Component | Value |
> | --- | --- |
> | Cost per NFE | 5.1629 s |
> | NFEs per latent section | 50 |
> | Time per latent section | 50 × 5.1629 = **258.15 s ≈ 4.30 min** |
> | Number of latent sections | 3 |
> | Time per stage for $\Phi_{\text{CF}}$ | 3 × 4.30 min = **12.9 min** |
>
> Thus, the Consistency-Guided Flow SDE takes approximately **12.9 minutes per stage**, or **25.8 minutes total** across the two stages.
>
> **Additional Overheads per Stage**
>
> For the sand castle example:
>
> - Stage 1 geometry processing (point-cloud unprojection, mesh reconstruction, volumetric sampling, segmentation): 7.3 minutes
> - Stage 2 MPM simulation & rendering (50,000 particles, 200 frames): 7.4 minutes (≈2.2 s per frame)
>
> **Total Runtime**
>
> Summing all components:
>
> - Stage 1 ($\Phi_{\text{CF}}$): 12.9 min
> - Stage 1 (geometry processing): 7.3 min
> - Stage 2 (simulation & render): 7.4 min
> - Stage 2 ($\Phi_{\text{CF}}$): 12.9 min
>
> Total: approximately 40 minutes end-to-end to generate a full physically realistic video from a single input image.
>
> **Contextualizing the Computational Cost**
>
> While alternative approaches achieve lower runtime, our method provides a full 3D scene representation that can be reused for subsequent simulation, editing, or multi-view generation. This capability is absent in both data-driven models (e.g., Sora [1], Veo3 [2], Gen-3 [3], FramePack [4]) and planning-based pipelines such as VLIPP [5], which do not produce persistent geometric assets.
> Importantly, our method is fully training-free. Systems that target similarly realistic simulation-to-video translation typically require extensive training. For example, Go-with-the-Flow [6] requires approximately 40 GPU-days using 8xA100 (80GB) GPUs to fine-tune CogVideoX-5B with warped noise. Compared to such training costs, our 40-minute inference time on a single consumer GPU represents a relatively modest computational cost for producing photorealistic, physically consistent videos along with reusable 3D assets.
> We will make these clarifications explicit in the revised manuscript.

---

> ### Author Response · Authors · 2025-11-25
> **Rebuttal by Authors**
>
> > **W3. Lack of limiation and discussion**
> >
>
> Thank you for the valuable comment. We will add our limitations and discussion regarding liquids in the revision, including the content mentioned in our reply to W1 and the content noted in our reply to W3 of another reviewer (evX1).
>
> ---
>
> > **Q1. Intermediate MPM output visualizations for different interaction types**
> >
>
> Please see the demo video Section 5. We have provided more examples and visualization of our MPM simulation outputs. Our method handles different types of collisions, shown in the demo video Section 1.
>
> ---
>
> **References**
>
> [1] Sora, OpenAI 2025.
>
> [2] Veo3, Google 2025.
>
> [3] Gen3, RunwayML 2024.
>
> [4] Frame Context Packing and Drift Prevention in Next-Frame-Prediction Video Diffusion Models, NeurIPS 2025.
>
> [5] VLIPP: Towards Physically Plausible Video Generation with Vision and Language Informed Physical Prior, ICCV 2025.
>
> [6] Go-with-the-Flow: Training-free Video Editing with Flow Matching, CVPR 2025.

---

### Official Review · Reviewer_DQZT · 2025-10-31

**Soundness:** 2
**Presentation:** 2
**Contribution:** 2
**Rating:** 2
**Confidence:** 4

**Summary:**

The paper presents 3DPhysVideo, a training-free pipeline designed to generate physically realistic videos from a single image input. It reuses a pre-trained Image-to-Video (I2V) flow model across two stages.

In Stage 1: Single Image to 3D, the I2V model functions as a view synthesizer to reconstruct 360-degree 3D scene geometry.

In Stage 2: Simulation to Video, Material Point Method (MPM) physics simulation is applied to the geometry. The resulting simulated point trajectories, which support complex dynamics like fluids and viscous substances, then guide the same I2V model to synthesize the final photorealistic video.

The core mechanism, Consistency-Guided Flow SDE, adapts the I2V model for both 3D reconstruction and simulation-guided rendering.

**Strengths:**

The 3DPhysVideo pipeline generates physically realistic videos from a single image using a training-free approach. It repurposes an off-the-shelf Image-to-Video (I2V) model in two stages.
1. 3D Reconstruction: The I2V model first acts as a novel view synthesizer to reconstruct 360-degree 3D scene geometry.
2. Physics Generation: The geometry undergoes Material Point Method (MPM) physics simulation. The resulting simulated dynamics then guide the same I2V model to synthesize the final photorealistic video.
This dual functionality is enabled by the Consistency-Guided Flow SDE, which adapts the pre-trained model for both geometry and dynamics synthesis. The method achieves good physical realism compared to baselines, especially in multi-object and fluid interaction scenarios, while offering user control over physical properties.

**Weaknesses:**

1. The proposed method appears incremental, with limited distinction from prior work.

2.Experiments are limited in scope; key baselines and datasets are missing.

3. Core assumptions lack rigorous justification or mathematical support.

4. Result interpretation is shallow; no discussion of failure cases or parameter sensitivity.

5. Figures and explanations are sometimes unclear, reducing readability and impact.

**Questions:**

1.	Could the authors elaborate on the empirical or theoretical rationale for entirely eliminating the denoising bias ?
2.	What is the measured reliability or accuracy of these automatically inferred physical parameters compared to manually specified inputs?
3.	Since the current SDE is heavily reliant on visual consistency, how would the core consistency metric and the model’s latent inputs need to be adapted or redefined to effectively enforce a non-visual inductive bias, such as alignment with a detailed text prompt, without requiring additional model training?

---

> ### Author Response · Authors · 2025-11-25
> **Rebuttal by Authors**
>
> We appreciate the reviewer’s thoughtful and detailed comments, and we are grateful for the positive recognition of our training-free pipeline, the dual use of the I2V model enabled by Consistency-Guided Flow SDE, and the strong physical realism and controllability demonstrated across challenging multi-object and fluid scenarios.
> The following provides our responses to the reviewer feedback, and an additional demo video has been newly included in the supplementary material.
>
> ---
>
> > **W1. Distinction from Prior Work**
> >
>
> We clarify the distinct differences between our method and prior work.
>
> For single image-to-3D reconstruction (Stage 1), the most relevant work PhysGen3D [1] and other prior works (e.g., PhysCtrl [2], PhysMotion [3]) adopt an object-centric 3D reconstruction approach. In contrast, our 3DPhysVideo takes a scene-level approach that reconstructs multiple objects simultaneously. This provides clear advantages by both preserving inter-object relationships and keeping reconstruction cost constant regardless of the number of objects. While there exist prior works that generate world-consistent 360-degree videos (e.g., DimensionX [4], Gen3C [5]), they require expensive fine-tuning. Thanks to our proposed Consistency-Guided Flow SDE $\Phi_{\text{CF}}$, to the best of our knowledge, we are the first to achieve world-consistent 360-degree video generation in a training-free manner by simply plugging in off-the-shelf video models.
>
> For simulation-to-video (Stage 2), PhysGen3D [1] uses Physically-Based Rendering, whereas we leverage the video model as a photorealistic renderer. While PhysMotion [3] also refines coarse videos using a video model, they first refine keyframes and then employ an additional TokenFlow [6] model to obtain the final video. VLIPP [7] uses Go-with-the-flow [8], a motion-guided video generation model, based on physics-grounded trajectories predicted by VLMs. However, as demonstrated in Section 2 of our demo video, the proposed $\Phi_{\text{CF}}$ enables off-the-shelf video models to directly operate like Go-with-the-Flow [8] in a training-free manner, without any expensive fine-tuning.
>
> ---
>
> > **W2. Experiments are limited in scope; key baselines and datasets are missing**
> >
>
> As mentioned in the experiments section of the paper, We already conducted comparisons with the most relevant and powerful baselines across various aspects, both for the total pipeline (Stage 1 + 2) and, even for simulation-to-video (Stage 2) only.
>
> **(1) Total pipeline (image-to-video)**
>
> - Pure data-driven image-to-video: Gen3 [9], Sora [10], Veo [11]
> - Explicit physics simulator (most relevant work): PhysGen3D [1]
> - Implicit physics using pre-trained model: VLIPP [7]
> - Direct baseline of our proposed method: FramePack [12]
>
> **(2) Stage 2 (simulation-to-video)**
>
> - Motion prior: MotionClone [13]
> - Flow-based: Go-with-the-Flow [8]
> - Mask-based: MagicMotion [14]
>
> Regarding datasets, following prior works (PhysGen3D [1], PhysMotion [2]), we evaluate on a curated dataset as is commonly done. However, unlike these prior works, we specifically constructed our dataset to cover diverse physical scenarios, including single-object, multi-object, and fluid dynamics, for more robust validation.
> Additionally, as shown in our demo video Section 1, we conducted further experiments on wind effects, cloth dynamics, and complex collision scenarios.

---

> ### Author Response · Authors · 2025-11-25
> **Rebuttal by Authors**
>
> > **W3 & Q1. Empirical or theoretical justification for entirely eliminating the denoising bias.**
> >
>
> We thank the reviewer for highlighting this point.
>
> Theoretically, in our main paper's Eq. 4, when $\beta \to \infty$, the KL divergence term is ignored, and the optimal distribution $p^{*}$ will not be able to maintain the original distribution $q = N((1-\tau)\mu, {\tau}^2 I)$. When $\beta \to 0$, the consistency term is ignored, so the original distribution is well preserved but consistency optimization does not occur. Therefore, a non-extreme choice of $\beta$ is required, and among these, our choice of $\beta = (1-\tau)/\tau$ is empirically the most practical.
>
> We verified this through additional experiments. For experiments with different $\beta$ values, we explicitly needed a additional denoising bias model $v_\epsilon$. To achieve this, we modified the existing $v_\theta(z_t, z_I, t)$ to remove the consistency bias by conditioning it on the first frame of the video latent $z_t$ instead of the image latent $z_I$ at every SDE optimization step, i.e., $v_\epsilon = v_\theta(z_t, z_t[0], t)$.
>
> From the experimental results in Section 3 of the demo video, when $\beta \to \infty$, the latent cannot maintain the original distribution, resulting in distorted videos. When $\beta \to 0$, consistency is not optimized, producing non-photorealistic videos.
> In contrast, within an appropriate range of $\beta$, the desired videos are generated, showing that our Consistency-Guided Flow SDE works well with low sensitivity to the hyperparameter $\beta$. However, among these $\beta$ , only our choice entirely eliminates $v_\epsilon$. As mentioned in our main paper, this enables our objective to be achieved using only the existing $v_\theta$, resulting in only half the number of function evaluations (NFEs), which is the most practical choice.
>
> ---
>
> > **W4. Discussion of limitations or parameter sensitivity**
> >
>
> Thank you for the valuable feedback.
>
> We will add a discussion on the limitation of volumetric liquids, as observed in the failure case in Section 7 of the demo video. In addition, we conducted additional experiments on the hyperparameters of our Consistency-Guided Flow SDE to adderess the feedback.
> The results of our experiments on the SDE target timestep $\tau$ and the parameter $\beta$, which was noted in W3 and Q1, have been added to Section 3 of the demo video.
> Both experiments show that our method remains stable and insensitive within a reasonably wide range of these parameters.
> Furthermore, to assess the sensitivity of the step size $\gamma$, which should be sufficient small to ensure the convergence of the Euler–Maruyama discretization, we monitored the norm of the optimized video latents at each SDE optimization step.
> This allows us to verify whether the original distribution is well preserved by the KL term in our objective Eq. 4.
> From the experiments, we observed that for $\gamma \le 0.8$, the optimization successfully converges.
> We will include in the revision the experimental results and discussion on parameter sensitivity, along with an analysis of the total number of SDE iterations $N$.
>
> ---
>
> > **W5. Figures and explanations are sometimes unclear, reducing readability and impact**
> >
>
> We thank the reviewer for the helpful feedback. To improve clarity and readability, we will revise the figures and explanations in the revision.

---

> ### Author Response · Authors · 2025-11-25
> **Rebuttal by Authors**
>
> > **Q2. Reliability and Accuracy concerns of automatically inferred phyiscal parameters**
> >
>
> We thank the reviewer for the helpful comment. We agree that automatic physical-property estimation can introduce errors, and inaccurate material parameters may lead to unrealistic simulation behavior. This issue is not specific to our pipeline but is inherent to current single-image physics-aware approaches. PhysGen3D [1] explicitly estimates elasticity and density using an VLM, VLIPP [7] relies on a vision-language model’s physical reasoning to infer material properties when planning motion trajectories, and recent works such as PhysT2V [15] follow a similar paradigm. These pipelines all depend on VLM based inference because such models currently provide the most practical and broadly applicable source of physical priors without requiring specialized datasets.
>
> There is also an emerging line of work, including Pixie [16], that focuses on predicting more accurate material properties from a single image. Integrating such methods into our pipeline could further improve stability, and we will clarify this direction in the revision.
>
> To assess the reliability of our physical-property estimation, we conducted an additional experiment on the Physics-IQ ball-drop scene by querying GPT ten times independently. Density predictions were stable, with a mean of 255 kg/m³, a standard deviation of 15, and a range of 250 to 300. Young’s modulus predictions had a mean of 5.75 × 10⁷, a standard deviation of 1.6 × 10⁷, and values between 5 × 10⁷ and 1 × 10⁸. Although the variance may seem large, Young’s modulus in real materials spans many orders of magnitude, so these predictions remain within reasonable bounds.
>
> ---
>
> > **Q3. What needs to be changed to use a non-visual inductive bias with the proposed SDE?**
> >
>
> As long as the initial video to be optimized is misaligned with the given detailed text, nothing needs to be changed. In Section A of the supplementary material, our decomposed consistency bias $v_c$ approximates $\nabla_{z_\tau}C(z_{\tau}, z_I)$, eliminating the need for an explicit consistency metric. Therefore, as mentioned in the conclusion of the paper, situations lacking text alignment will reveal the text alignment bias $v_{text}$ that the model learned during training. In the same way as $\nabla_{z_\tau}C(z_{\tau}, z_I) \approx v_c = v_{\theta} - v_{\epsilon}$ in our task, this $v_{text} = v_{\theta} - v_{\epsilon}$ approximates $\nabla_{z_\tau}\text{TextAlignment}(z_{\tau}, z_{text})$ (where $z_{text}$ is the latent of the text feature) and serves as a component of the score term in the SDE for text alignment optimization.
>
> Section 2 of our demo video demonstrates the results of applying our SDE with text alignment inductive bias.
>
> Additionally, the masked inversion strategy for the model's latent inputs is used to preserve specific regions for our task. For other tasks that do not require preserving specific regions, it can simply be replaced with the forward process.
> We will add experimental results and implementation details in the revision.
>
> ---
>
> **References**
>
> [1] PhysGen3D: Crafting a Miniature Interactive World from a Single Image, CVPR 2025.
>
> [2] PhysCtrl: Generative Physics for Controllable and Physics-Grounded Video Generation, NeurIPS 2025.
>
> [3] PhysMotion: Physics-Grounded Dynamics from a Single Image, ArXiv 2024.
>
> [4] DimensionX: Create Any 3D and 4D Scenes from a Single Image with Decoupled Video Diffusion, ICCV 2025.
>
> [5] GEN3C: 3D-Informed World-Consistent Video Generation with Precise Camera Control, CVPR 2025.
>
> [6] TokenFlow: Consistent Diffusion Features for Consistent Video Editing, ICLR 2024.
>
> [7] VLIPP: Towards Physically Plausible Video Generation with Vision and Language Informed Physical Prior, ICCV 2025.
>
> [8] Go-with-the-Flow: Training-free Video Editing with Flow Matching, CVPR 2025.
>
> [9] Gen3, RunwayML 2024.
>
> [10] Sora, OpenAI 2025.
>
> [11] Veo3, Google 2025.
>
> [12] Frame Context Packing and Drift Prevention in Next-Frame-Prediction Video Diffusion Models, NeurIPS 2025.
>
> [13] Motionclone: Training-free motion cloning for controllable video generation, ICLR 2025.
>
> [14] Magicmotion: Controllable video generation with dense-to-sparse trajectory guidance,  ICCV 2025.
>
> [15] PhyT2V: LLM-Guided Iterative Self-Refinement for Physics-Grounded Text-to-Video Generation, CVPR 2025.
>
> [16] Pixie: Fast and Generalizable Supervised Learning of 3D Physics from Pixels, ArXiv 2025.

---

> ### Author Response · Authors · 2025-12-02
> **Rebuttal by Authors**
>
> > **W3 & Q1. Empirical or theoretical justification for entirely eliminating the denoising bias.**
> >
>
> In the previous reply, we focused on **why we set $\beta=\frac{1-\tau}{\tau}$ for our final iterative formula (Eq. 5) so that the denoising bias term $v_{\epsilon}$ is entirely eliminated**. Since it was not possible to discuss with the reviewer and clearly identify the precise meaning of the entirely eliminating denoising bias, we provide additional clarification regarding this concern for completeness.
>
> In this reply, we clarify **why we entirely eliminate the denoising bias to optimize the initial inconsistent latent $z_\tau$ into the consistency-optimal video latent $z_{\tau}^{\*}$**. In other words, we explain why we perform optimization without any denoising, so that the latent stays in the original distribution at diffusion timestep $t=\tau$, and then apply only a single final denoising process at the end ($z_\tau\to$ SDE $\to z_{\tau}^{\*} \to$ Final denoising $\to z^{\*}$).
>
> First, the case where we do not remove any denoising bias, which corresponds to the standard generation process, has already been discussed in the paper. As shown in the SDE steps ablation figure, this produces consistency-suboptimal videos.
> Instead, if we partially allow the denoising bias to guide the optimization (e.g., $z_\tau\to$ SDE $\to$ Denoising $\to z_{\tau-1} \to$ SDE $\to$ Denoising $\to z_{\tau-2} \ldots \to$ Remaining denoising $\to z^*$), the original latent distribution at $t=\tau$ gradually shifts toward a less noisy distribution with each denoising step.
> As we show in the $\tau$-ablation results in Section 3 of our demo video, as the latent shift toward a less noisy distribution (i.e., as $t \to 0$), the model focuses more on high-frequency details, and can no longer enforce consistency on low-frequency components.
> Therefore, if denoising is applied before the latent is sufficiently optimized at the timestep where the desired consistency can be enforced (in our setting, $\tau$-index $= 20$), the model cannot properly enforce the desired consistency.
> Also, if the latent is already optimized enough at that timestep, we empirically observed that subsequent SDE optimization after denoising in the less noisy distribution brings no significant improvement, since the latent is already on the consistency-optimal flow ODE path of the I2V flow model prior. For these reasons, it is reasonable and practical to entirely eliminate the denoising bias, keep the latent at the timestep where consistency can be properly enforced, perform SDE optimization there, and then apply a single final denoising at the end.

---

### Official Review · Reviewer_evX1 · 2025-10-31

**Soundness:** 4
**Presentation:** 3
**Contribution:** 3
**Rating:** 8
**Confidence:** 5

**Summary:**

This paper introduces 3DPhysVideo, a novel, training-free pipeline that generates physically realistic videos from a single input image. Instead of training a new model, it cleverly repurposes a single, pre-trained image-to-video (I2V) model for two distinct stages: 3D Scene Reconstruction and Physics-Guided Video Generation.

**Strengths:**

1. The entire pipeline requires no additional training. It runs on a single consumer GPU, making it highly accessible and efficient compared to methods that require training large, specialized models.
2. By grounding the animation in an explicit physics engine (MPM), the final video exhibits a high degree of physical plausibility, especially in complex scenarios like fluid dynamics and multi-object interactions, where purely data-driven models (e.g., Sora, Gen-3) often fail.
3. The paper is well written and organized.

**Weaknesses:**

1. As a multi-stage pipeline, errors from any stage may make the result fail. In particular, the 3D reconstruction and physical property estimation (using LLM) parts are prone to errors. For example, the apple in the demo appears elastic (it should actually be similar to a rigid body). It would be better if the accuracy of these two parts could be assessed, and the potential limitations could be analyzed.
2. While it can run on a consumer GPU, this method predictably significantly increases inference time due to the introduction of 3D reconstruction, physical property estimation, and MPM simulation. It would be better to report a comparison of inference time.
3. Were the liquids in the scene also reconstructed in 3D? How is the physical realism of the fluid dynamics ensured?
4. The article states that PhysGen3D cannot maintain the relative position of objects. However, PhysGen3D does perform pose estimation, so is this statement somewhat unreasonable?

**Questions:**

Please see Weaknesses.

---

> ### Author Response · Authors · 2025-11-25
> **Rebuttal by Authors**
>
> We appreciate the reviewer’s helpful and encouraging comments, especially the recognition of our training-free, single-GPU pipeline and the strengths of grounding video generation in explicit physics simulation, as well as the positive feedback on the clarity and organization of the paper.
> The following provides our responses to the reviewer feedback, and an additional demo video has been newly included in the supplementary material.
>
> ---
>
> > **W1. Reliability issue with VLM inferred physical properties & 3D reconstruction.**
> >
>
> We thank the reviewer for the helpful comment. We agree that both 3D reconstruction and automatic physical parameter estimation can introduce errors. For the 3D reconstruction, we acknowledge that there are some cases where 3D reconstruction fails, especially when dealing with volumetric liquids. The failure case is shown in Section 7 of demo video.
>
>  Also, inaccurate VLM inferred physical parameters may lead to unrealistic simulation behavior. This issue is not specific to our pipeline but is inherent to current single-image physics-aware approaches. PhysGen3D [1] explicitly estimates elasticity and density using an VLM, VLIPP [2] relies on a vision-language model’s physical reasoning to infer material properties when planning motion trajectories, and recent works such as PhyT2V [3] follow a similar paradigm. These pipelines all depend on VLM based inference because such models currently provide the most practical and broadly applicable source of physical priors without requiring specialized datasets.
>
> There is also an emerging line of work, including Pixie [4], that focuses on predicting more accurate material properties from a single image. Integrating such methods into our pipeline could further improve stability, and we will clarify this direction in the revision.
> To assess the reliability of our physical-property estimation, we conducted an additional experiment on the Physics-IQ ball-drop scene by querying GPT ten times independently. Density predictions were stable, with a mean of 255 kg/m³, a standard deviation of 15, and a range of 250 to 300. Young’s modulus predictions had a mean of 5.75 × 10⁷, a standard deviation of 1.6 × 10⁷, and values between 5 × 10⁷ and 1 × 10⁸. Although the variance may seem large, Young’s modulus in real materials spans many orders of magnitude, so these predictions remain within reasonable bounds.

---

> ### Author Response · Authors · 2025-11-25
> **Rebuttal by Authors**
>
> > **W2. Inference Time analysis**
> >
>
> We thank the reviewer for the question regarding the runtime distribution across the stages of our pipeline. Below, we provide a complete inference-time analysis using a single NVIDIA RTX 3090 GPU. The pipeline consists of two major stages : Single Image to 3D (Stage 1) and Simulation to Video (Stage 2). Each stage includes a Consistency-Guided Flow SDE step.
>
> A video is processed in three latent sections, where each latent section generates 36 frames. Each latent section requires 50 NFEs (20 inversion, 10 SDE refinement, and 20 final denoising). The measured cost per NFE is 5.1629 seconds.
>
> **NFE and Latent Section Cost**
>
> | Component | Value |
> | --- | --- |
> | Cost per NFE | 5.1629 s |
> | NFEs per latent section | 50 |
> | Time per latent section | 50 × 5.1629 = **258.15 s ≈ 4.30 min** |
> | Number of latent sections | 3 |
> | Time per stage for $\Phi_{\text{CF}}$ | 3 × 4.30 min = **12.9 min** |
>
> Thus, the Consistency-Guided Flow SDE takes approximately **12.9 minutes per stage**, or **25.8 minutes total** across the two stages.
>
> **Additional Overheads per Stage**
>
> For the sand castle example:
>
> - Stage 1 geometry processing (point-cloud unprojection, mesh reconstruction, volumetric sampling, segmentation): 7.3 minutes
> - Stage 2 MPM simulation & rendering (50,000 particles, 200 frames): 7.4 minutes (≈2.2 s per frame)
>
> **Total Runtime**
>
> Summing all components:
>
> - Stage 1 ($\Phi_{\text{CF}}$): 12.9 min
> - Stage 1 (geometry processing): 7.3 min
> - Stage 2 (simulation & render): 7.4 min
> - Stage 2 ($\Phi_{\text{CF}}$): 12.9 min
>
> Total: approximately 40 minutes end-to-end to generate a full physically realistic video from a single input image.
>
> **Contextualizing the Computational Cost**
>
> While alternative approaches achieve lower runtime, our method provides a full 3D scene representation that can be reused for subsequent simulation, editing, or multi-view generation. This capability is absent in both data-driven models (e.g., Sora [5], Veo3 [6], Gen-3 [7], FramePack [8]) and planning-based pipelines such as VLIPP [2], which do not produce persistent geometric assets.
>
> Importantly, our method is fully training-free. Systems that target similarly realistic simulation-to-video translation typically require extensive training. For example, Go-with-the-Flow [9] requires approximately 40 GPU-days using 8xA100 (80GB) GPUs to fine-tune CogVideoX-5B with warped noise. Compared to such training costs, our 40-minute inference time on a single consumer GPU represents a relatively modest computational cost for producing photorealistic, physically consistent videos along with reusable 3D assets.
>
> We will make these clarifications explicit in the revised manuscript.

---

> ### Author Response · Authors · 2025-11-25
> **Rebuttal by Authors**
>
> > **W3. 3D reconstruction of liquids and physical realism of fluid dynamics**
> >
>
> input simulation videos in Figure 4 of the paper and Section 5 of demo video. These liquid point clouds are simulated using Newtonian fluid.  When objects fall, the points spread like liquid, and this simulated motion serves as guidance for the video model's prior to generate more realistic fluid dynamics.
>
> However, while our pipeline works for shallow liquids on the ground (as in our scenarios), it has a limitation when dealing with volumetric liquids such as water in cups or lake scenes, where accurate 3D reconstruction becomes much more difficult. Failure case while processing volumentric water is shown in Section 7 of our demo video.
>
> To explore a potential direction for alleviating this limitation, we conducted additional experiments applying our Consistency-Guided Flow SDE $\Phi_{\text{CF}}$  to more generic scenarios. Inspired by VLIPP [2], we leverage physics-grounded motion inferred from VLMs and apply our $\Phi_{\text{CF}}$ to guide the video generation with this motion in a manner similar to Go-with-the-Flow [9], while remaining entirely training-free. Our experiments demonstrate that this approach can handle volumetric liquids and more diverse scenes in a more efficient manner. We will include both the pipeline limitation and these additional application results in the revision.
>
> ---
>
> > **W4. Relative position accuracy compared to PhysGen3D**
> >
>
> We appreciate the reviewer’s question. While PhysGen3D [1] performs relative pose estimation, it first separates the scene into individual objects, reconstructs each object independently from its own cropped image, and then reassembles them. Because this process relies on object-wise reconstruction, it can fail in scenes with occlusion or tightly coupled interactions, where relative positions must be inferred jointly.
>
> Our method reconstructs the entire scene at once, before any masking, so the spatial relationships between objects are naturally preserved. In practice, PhysGen3D [1] failed on our “ball colliding with dominos” example (demo video Section 1), where pose estimation produced an error and no valid output was generated. Section 4 of demo video and first scenario in Figure 3 of main paper shows that our method accurately places object in the scenes where PhysGen3D fail.
>
> ---
>
> **References**
>
> [1] PhysGen3D: Crafting a Miniature Interactive World from a Single Image, CVPR 2025.
>
> [2] VLIPP: Towards Physically Plausible Video Generation with Vision and Language Informed Physical Prior, ICCV 2025.
>
> [3] PhyT2V: LLM-Guided Iterative Self-Refinement for Physics-Grounded Text-to-Video Generation, CVPR 2025.
>
> [4] Pixie: Fast and Generalizable Supervised Learning of 3D Physics from Pixels, ArXiv 2025.
>
> [5] Sora, OpenAI 2025.
>
> [6] Veo3, Google 2025.
>
> [7] Gen3, RunwayML 2024.
>
> [8] Frame Context Packing and Drift Prevention in Next-Frame-Prediction Video Diffusion Models, NeurIPS 2025.
>
> [9] Go-with-the-Flow: Training-free Video Editing with Flow Matching, CVPR 2025.

---

### Official Review · Reviewer_heD9 · 2025-11-01

**Soundness:** 3
**Presentation:** 3
**Contribution:** 3
**Rating:** 4
**Confidence:** 3

**Summary:**

The paper introduces 3DPhys Video, a training-free pipeline for generating physically realistic and photorealistic videos from a single input image. It addresses the fundamental limitation of traditional video generative models, which often fail to adhere to real-world physical dynamics.

The pipeline operates in two main stages, Novel View Synthesis and Simulation to Video Generation.

**Strengths:**

The core contribution is the training-free pipeline that repurposes an off-the-shelf image-to-video diffusion model for two entirely different tasks: 3D scene reconstruction and physics-guided video synthesis.

The authors conducted extensive experiments to validate the effectiveness of their proposed method. The results demonstrate that 3DPhysVideo outperforms state-of-the-art methods in terms of physical realism and semantic consistency while maintaining competitive photorealism.

**Weaknesses:**

The Material Point Method (MPM) is computationally expensive, especially for high-resolution simulations and complex scenes with numerous interaction points. The overall pipeline's speed is likely bottlenecked by the MPM step. The authors should clearly address the runtime breakdown for the three main stages: 3D reconstruction, MPM simulation, and I2V synthesis, to highlight the practical efficiency of the "training-free" claim.

The demonstration mostly focuses on relatively contained scenes with specific, localized physical events (e.g., ball drops, liquid pouring). It is unclear how well the pipeline scales to large-scale, non-local physical phenomena like wind effects, cloth dynamics, or complex collisions involving many small particles.

**Questions:**

Could you address the problems in the weaknesses?

---

> ### Author Response · Authors · 2025-11-25
> **Rebuttal by Authors**
>
> We appreciate the reviewer’s helpful and insightful comments, as well as the positive recognition of our training-free pipeline and the strength of our experimental results in physical realism, semantic consistency, and photorealism.
> The following provides our responses to the reviewer feedback, and an additional demo video has been newly included in the supplementary material.
>
> ---
>
> > **W1. Inference Time analysis**
> >
>
> We thank the reviewer for the question regarding the runtime distribution across the stages of our pipeline. Below, we provide a complete inference-time analysis using a single NVIDIA RTX 3090 GPU. The pipeline consists of two major stages : Single Image to 3D (Stage 1) and Simulation to Video (Stage 2). Each stage includes a Consistency-Guided Flow SDE step.
>
> A video is processed in three latent sections, where each latent section generates 36 frames. Each latent section requires 50 NFEs (20 inversion, 10 SDE refinement, and 20 final denoising). The measured cost per NFE is 5.1629 seconds.
>
> **NFE and Latent Section Cost**
>
> | Component | Value |
> | --- | --- |
> | Cost per NFE | 5.1629 s |
> | NFEs per latent section | 50 |
> | Time per latent section | 50 × 5.1629 = **258.15 s ≈ 4.30 min** |
> | Number of latent sections | 3 |
> | Time per stage for $\Phi_{\text{CF}}$ | 3 × 4.30 min = **12.9 min** |
>
> Thus, the Consistency-Guided Flow SDE takes approximately **12.9 minutes per stage**, or **25.8 minutes total** across the two stages.
>
> **Additional Overheads per Stage**
>
> For the sand castle example:
>
> - Stage 1 geometry processing (point-cloud unprojection, mesh reconstruction, volumetric sampling, segmentation): 7.3 minutes
> - Stage 2 MPM simulation & rendering (50,000 particles, 200 frames): 7.4 minutes (≈2.2 s per frame)
>
> **Total Runtime**
>
> Summing all components:
>
> - Stage 1 ($\Phi_{\text{CF}}$): 12.9 min
> - Stage 1 (geometry processing): 7.3 min
> - Stage 2 (simulation & render): 7.4 min
> - Stage 2 ($\Phi_{\text{CF}}$): 12.9 min
>
> Total: approximately 40 minutes end-to-end to generate a full physically realistic video from a single input image.
>
> **Contextualizing the Computational Cost**
>
> While alternative approaches achieve lower runtime, our method provides a full 3D scene representation that can be reused for subsequent simulation, editing, or multi-view generation. This capability is absent in both data-driven models (e.g., Sora [1], Veo3 [2], Gen-3 [3], FramePack [4]) and planning-based pipelines such as VLIPP [5], which do not produce persistent geometric assets.
> Importantly, our method is fully training-free. Systems that target similarly realistic simulation-to-video translation typically require extensive training. For example, Go-with-the-Flow [10] requires approximately 40 GPU-days using 8xA100 (80GB) GPUs to fine-tune CogVideoX-5B with warped noise. Compared to such training costs, our 40-minute inference time on a single consumer GPU represents a relatively modest computational cost for producing photorealistic, physically consistent videos along with reusable 3D assets.
>
> We will make these clarifications explicit in the revised manuscript.

---

> ### Author Response · Authors · 2025-11-25
> **Rebuttal by Authors**
>
> > **W2. Scalability to Large-Scale and Non-Local Physical Events(wind effcts, cloth dynamics, complex collisions)**
> >
>
> We conducted additional experiments on scenes that include wind effects, cloth dynamics, and complex collisions. The results are shown in Section 1 of our demo video.
>
> For wind effects, we note that our MPM simulator can handle such phenomena directly, as was done in Pixie [6].
> For cloth dynamics, we adjusted physical parameters such as Young’s modulus to match real clothing materials, which makes it possible to simulate items like socks and towels to a certain degree.
> For more detailed cloth behavior, anisotropic MPM (as used in MPMAvatar [7]) can be used, and it can be plugged into our pipeline.
> For complex collisions, we set up a new dominoes scene where a ball hits the stack. This demonstrates that our pipeline can handle complex collisions and scale to larger scenes, since our method has no limits on the number or size of objects.
>
> As W1 pointed out, the need for high-resolution MPM simulations in large-scale scenes leads to higher computational cost.
> Nevertheless, at least in the simulation preparation stage, our scene-level reconstruction approach improves scalability than prior works such as PhysGen3D [8] and PhysMotion [9], which rely on object-level 3D reconstruction.
>
> As a potential direction to further improve scalability, we could replace the MPM simulator with a physics-reasoning approach, similar to VLIPP [5]. Even in this case, our Consistency-Guided Flow SDE $\Phi_{\text{CF}}$ enables off-the-shelf video models to operate as Go-with-the-Flow [10] in a training-free manner. These results are shown in section 2 of demo video. This opens up opportunities for large-scale and more general scenes.
>
> ---
>
> **References**
>
> [1] Sora, OpenAI 2025.
>
> [2] Veo3, Google 2025.
>
> [3] Gen3, RunwayML 2024.
>
> [4] Frame Context Packing and Drift Prevention in Next-Frame-Prediction Video Diffusion Models, NeurIPS 2025.
>
> [5] VLIPP: Towards Physically Plausible Video Generation with Vision and Language Informed Physical Prior, ICCV 2025.
>
> [6] Pixie: Fast and Generalizable Supervised Learning of 3D Physics from Pixels, ArXiv, 2025.
>
> [7] MPMAvatar: Learning 3D Gaussian Avatars with Accurate and Robust Physics-Based Dynamics, NeurIPS, 2025.
>
> [8] PhysGen3D: Crafting a Miniature Interactive World from a Single Image, CVPR 2025.
>
> [9] Physmotion: Physics-grounded dynamics from a single image. arXiv, 2024.
>
> [10] Go-with-the-Flow: Training-free Video Editing with Flow Matching, CVPR 2025.

---

### Author Response · Authors · 2025-12-03

We sincerely thank the reviewers and the AC for the time and care they dedicated to evaluating our work. We have revised the manuscript by incorporating the recommended additional experiments and by further improving the clarity of the writing and figures (highlighted in blue in the PDF). We also updated the supplementary video to include more qualitative comparisons. We appreciate the valuable feedback from the reviewers and the AC.

---

### Note · Program_Chairs · 2026-01-17
**Submission Desk Rejected by Program Chairs**

The following references in this submission do not refer to real documents and/or have major errors in bibliographic information:

 Yoonwoo Jeong, Sunghwan Kim, Sanghyun Ryu, Gihyun Kwon, Youngjung Uh, Youngjoon Kim, and Junmo Joo. Track4gen: Teaching video diffusion models to track points improves video generation. In CVPR, 2025.